# Regulation of mycobacterial infection by macrophage *Gch1* and tetrahydrobiopterin

Eileen McNeill [1,2], Elena Stylianou [3], Mark J. Crabtree [1,2], Rachel Harrington-Kandt[3], Anna-Lena Kolb[1,2], Marina Diotallevi[1,2], Ashley B. Hale[1,2], Paulo Bettencourt[3], Rachel Tanner[3], Matthew K. O'Shea[3], Magali Matsumiya[3], Helen Lockstone[2], Julius Müller [3], Helen A. Fletcher [4], David R. Greaves[5], Helen McShane[3] & Keith M. Channon[1,2]

Inducible nitric oxide synthase (iNOS) plays a crucial role in controlling growth of *Mycobacterium tuberculosis* (*M.tb*), presumably via nitric oxide (NO) mediated killing. Here we show that leukocyte-specific deficiency of NO production, through targeted loss of the iNOS cofactor tetrahydrobiopterin (BH4), results in enhanced control of *M.tb* infection; by contrast, loss of iNOS renders mice susceptible to *M.tb*. By comparing two complementary NO-deficient models, *Nos2*$^{-/-}$ mice and BH4 deficient *Gch1*$^{fl/fl}$Tie2cre mice, we uncover NO-independent mechanisms of anti-mycobacterial immunity. In both murine and human leukocytes, decreased *Gch1* expression correlates with enhanced cell-intrinsic control of mycobacterial infection in vitro. Gene expression analysis reveals that *Gch1* deficient macrophages have altered inflammatory response, lysosomal function, cell survival and cellular metabolism, thereby enhancing the control of bacterial infection. Our data thus highlight the importance of the NO-independent functions of *Nos2* and *Gch1* in mycobacterial control.

[1] Division of Cardiovascular Medicine, Radcliffe Department of Medicine, John Radcliffe Hospital, University of Oxford, Oxford OX3 9DU, UK. [2] Wellcome Trust Centre for Human Genetics, University of Oxford, Oxford OX3 7BN, UK. [3] Jenner Institute, University of Oxford, Oxford OX3 7DQ, UK. [4] Department of Immunology and Infection, London School of Hygiene and Tropical Medicine, Keppel Street, London WC1E 7HT, UK. [5] Sir William Dunn School of Pathology, University of Oxford, Oxford OX1 3RE, UK. These authors contributed equally: Eileen McNeill, Elena Stylianou. These authors jointly supervised this work: Helen McShane, Keith M. Channon. Correspondence and requests for materials should be addressed to E.M. (email: eileen.mcneill@well.ox.ac.uk) or to K.M.C. (email: keith.channon@cardiov.ox.ac.uk)

Tuberculosis (TB) remains a significant global health problem that kills more people than any other infectious disease, with>10 million new cases and 1.8 million deaths annually[1]. Chronic HIV co-infection and the treatment of chronic inflammatory conditions such as rheumatoid arthritis with potent immune modulators such as anti-TNF biologics, has revealed latent *Mycobacterium tuberculosis* (*M.tb*) infection reactivation as another growing source of TB disease burden, compounded by the global rise of drug-resistant TB. Thus, the discovery of new treatments for TB remains a pressing priority. In particular, augmenting host anti-mycobacterial mechanisms to enhance pathogen control through host-directed therapies (HDTs) is an area of intense interest. Identification of novel innate pathways and mechanisms could yield important insights that would lead to new therapeutic strategies.

Progress in the identification of genetic factors involved in TB susceptibility has been limited but the importance of some genes is already clear[2]. Polymorphisms in inducible nitric oxide synthase (iNOS, encoded by *NOS2*)[3] confer important differences in the control of mycobacterial infection. Mice deficient in iNOS show variability in the timing of loss of control of mycobacterial infection in experimental models, however, in all published studies to date $Nos2^{-/-}$ mice show increased susceptibility at later time points. $Nos2^{-/-}$ mice succumb to infection with intravenous *Mycobacterium bovis* Bacillus Calmette Guerin (BCG) after 8–12 weeks, in contrast to wildtype mice that are able to control infection[4]. Furthermore, iNOS-deficient mice infected with the more virulent *M.tb* Erdman succumb more rapidly than control mice in both intravenous and aerosol challenge experiments, even if early control of infection is maintained in short challenges[5–7]. Thus, iNOS is a pivotal regulator of susceptibility to *M.tb* infection in mice. These effects of iNOS have been associated with the production of nitric oxide (NO), largely because of in vitro correlations between the mycobacteriocidal activity of activated murine macrophages and the amount of reactive nitrogen intermediates (RNI) produced[8,9] and the direct cytotoxicity of nitric oxide to mycobacteria[10].

The concept that NO, and thus iNOS, acts only to kill mycobacteria in infected macrophages overlooks the more recent advances in the understanding of NO and NOS-dependent nitroso-redox signaling. NO can mediate reduced susceptibility to infection by modulating a range of host cell functions including induction of cell apoptosis or necrosis, control of phagosome maturation, effects on iron metabolism and signaling to the adaptive immune system[11]. Many of these functions are mediated by nitric oxide and RNI as signaling molecules, transducing their signal through protein modifications such as nitrosylation of cysteine residues or protein nitration. However, iNOS may also have nitric oxide-independent functions. The NOS cofactor tetrahydrobiopterin (BH4) is essential for NO production and stabilizes iNOS dimer formation[12]. Under BH4 replete conditions the NOS isoforms generate NO by oxidation of the amino acid L-arginine by molecular oxygen, forming L-citrulline[13]. NOS enzymes function in an 'uncoupled' state in the absence of BH4, whereby L-arginine is not oxidized to NO and superoxide is produced by the reductase domain of the enzyme. Whilst most attention on NOS uncoupling has been focused on endothelial NOS[14,15], we have demonstrated more recently that macrophage BH4 regulates production of reactive oxygen species, and impacts nitroso-redox dependent gene expression[16]. iNOS also demonstrates direct protein–protein interactions with a number of proteins including those with ribosomal and metabolic functions[17] indicating that the ability of this protein to influence cell biology is much wider than through NO-production alone.

The human biology of iNOS and NO in tuberculosis is controversial[18–20]. Although detection of functional iNOS in patients with inflammatory conditions such as tuberculosis has been documented[21,22], the ability of human macrophages to produce nitric oxide in vitro is less clear. However, an often overlooked aspect of human macrophage biology is their limited BH4 content[23]. How macrophage iNOS functions under conditions of limiting BH4 supply serves as a unique model of human infection, and alterations in disease processes under these conditions may reveal findings of particular relevance to human therapeutics in comparison to experimental models in wildtype mice where a fulminant NO response is seen.

BH4 is produced by a multistep pathway involving three enzymes; GTP cyclohydrolase-1 (GTPCH-1), pyrovoyl tetrahydropterin synthase (PTPS) and sepiapterin reductase (SR)[23]. GTPCH-1, encoded by *Gch1*, is involved in the initial and rate-limiting step of BH4 synthesis[24]. We have generated a conditional knockout mouse with *Gch1* deletion in leukocytes, preventing BH4 biosynthesis[25].

Here we demonstrate that mice deficient in NO production, through either loss of iNOS protein or the essential NOS co-factor BH4 show divergent phenotypes of susceptibility vs enhanced control of mycobacterial infection both in vivo and in vitro. Using genomic analysis, we identify a gene signature, unique to the loss of BH4 and distinct from the loss of iNOS protein, that is associated with enhanced control of mycobacterial infection. These results demonstrate that NO-independent mechanisms of bacterial control by iNOS and BH4 have pathological importance and therapeutic potential in human disease.

## Results

### iNOS$^+$ *Gch1$^{fl/fl}$*Tie2cre macrophages do not produce NO.

We generated matched litters of *Gch1$^{fl/fl}$*Tie2cre and *Gch1$^{fl/fl}$* mice (hereafter referred to as wildtype), and confirmed that *Gch1$^{fl/fl}$*Tie2cre mice lack *Gch1* expression in hematopoietic cells (Supplementary Figure 1a–c). In vitro infection of macrophages with BCG (Pasteur) alone showed minimal induction of *Nos2*, however in the presence of IFNγ infection with BCG caused a robust induction of *Nos2* and accumulation of nitrite, indicating robust iNOS NO-producing activity (Fig. 1a). The specificity of nitrite accumulation for iNOS activity was confirmed by infection of $Nos2^{-/-}$ BMDM under identical conditions (Supplementary Figure 2). BCG/IFNγ treatment caused increased cellular BH4 content in wildtype mice, with BH4 levels and GTPCH protein significantly reduced to near undetectable levels in the *Gch1$^{fl/fl}$*Tie2cre cells (Figure 1b, c). Loss of BH4 did not affect induction of iNOS protein (Fig. 1b), but prevented iNOS NO-producing activity measured as nitrite accumulation, arginine–citrulline conversion and direct detection of authentic NO generation by EPR (Fig. 1d–f). In order to test whether iNOS protein had other non-NO-producing roles in mycobacterial function the BCG/IFNγ combination was used in all in vitro studies to cause robust iNOS expression. These data confirm that whilst iNOS protein is induced in macrophages infected with BCG, in the absence of the co-factor BH4 no nitric oxide is produced, providing a system to discriminate iNOS functions reliant specifically on nitric oxide production.

Whilst nitric oxide production was absent, production of ROS was significantly increased in *Gch1$^{fl/fl}$*Tie2cre macrophages both at baseline (1.6 fold) and following BCG infection (3.2 fold) when assessed by the production of ethidium from dihydroethidium (Fig. 1g). This indicates that the loss of BH4 causes alterations in cellular redox state in both the presence and absence of iNOS. Macrophages have multiple potential sources of ROS, including the phagocytic NADPH oxidase that mediate oxidative burst and mitochondrial metabolism. To assess more broadly how ROS production by infected macrophages was affected, the phagocytic oxidative burst was assessed by flow cytometry using the CellROX

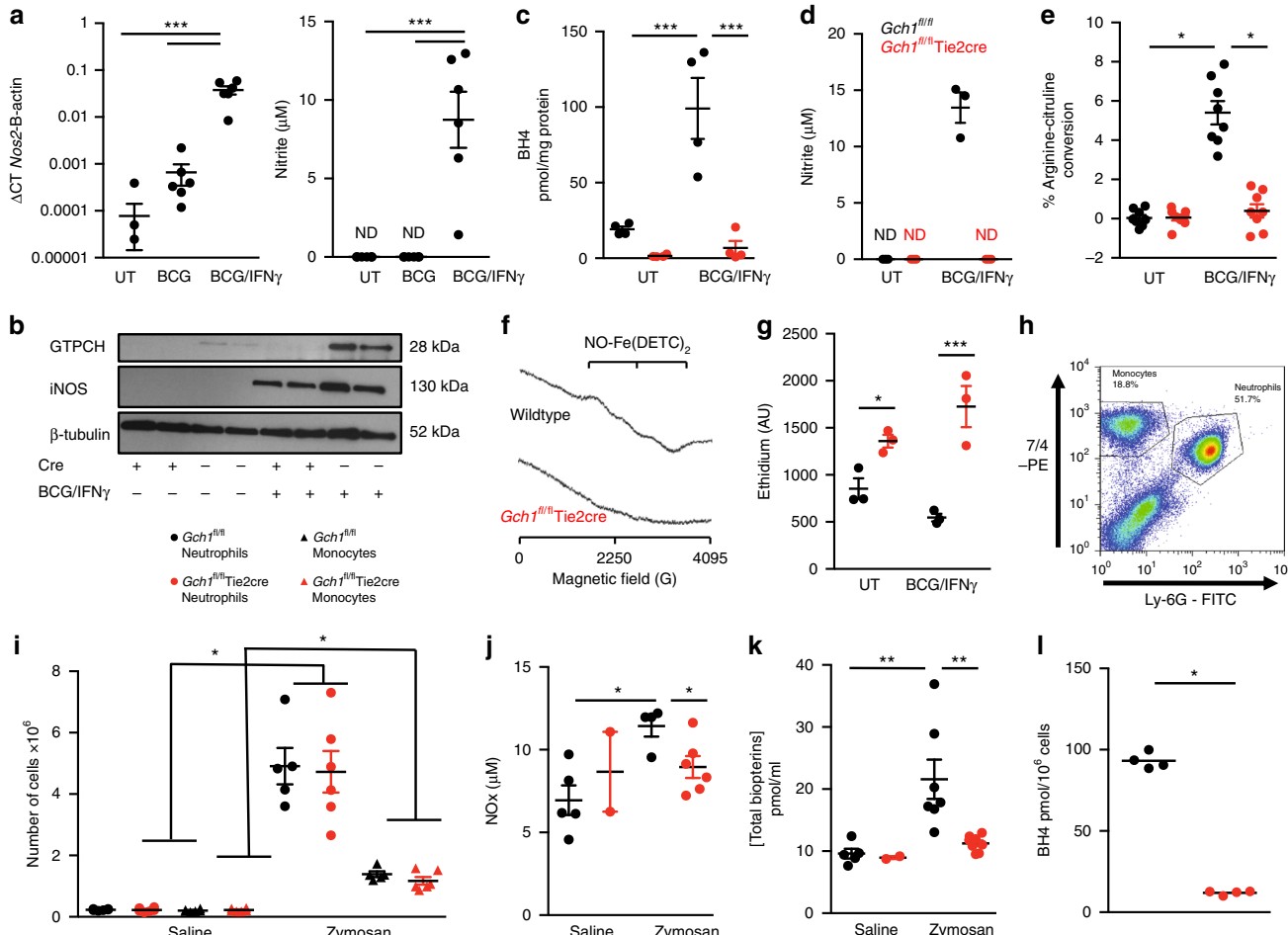

**Fig. 1** Characterisation of the in vitro mycobacterial infection of $Gch1^{fl/fl}$ Tie2cre macrophages. **a** Measurement of Nos2 and nitrite accumulation in bone marrow-derived macrophages stimulated with BCG (MOI 1:1) or IFNγ (10 ng/ml) and BCG (MOI 1:1) for 24 h in wildtype macrophages ($n = 6$). **b** Western blotting showing GTPCH and iNOS protein expression in $Gch1^{fl/fl}$Tie2cre macrophages, compared to wildtype (WT) macrophages at baseline and following infection with IFNγ (10 ng/ml) and BCG (MOI 1:1) for 24 h. Equal protein loading was demonstrated by detection of β-tubulin. **c** Measurement of BH4 (pmol mg$^{-1}$ protein) in bone marrow-derived macrophages ($n = 4$, $p < 0.05$). UT uninfected. **d** Nitrite accumulation in the cell culture media over 24 h of cell stimulation with BCG/IFNγ as before was measured using the Griess assay ($n = 3$/genotype). ND not detectable. **e** Quantification of the NMA-inhibitable arginine to citrulline conversion ($n = 8$, $p < 0.05$). **f** Representative EPR spectra from $n = 4$/genotype WT and $Gch1^{fl/fl}$Tie2cre macrophages following activation with IFNγ (10 ng/ml) and BCG (MOI 1:1). **g** Cells were incubated with dihydroethidium for the last 30 min of culture and ethidium production was measured in cell lysates by HPLC ($n = 3$/genotype). **h** Flow cytometry dotplot demonstrating identification of neutrophils (7/4$^{HI}$, Ly-6G$^{HI}$) and inflammatory monocytes (7/4$^{HI}$, Ly-6G$^{-}$) in lavage fluid from wildtype animals 6 h after injection with 1 mg zymosan ip. **i** The recruitment of monocytes and neutrophils by zymosan or saline control injection into wildtype and $Gch1^{fl/fl}$Tie2cre animals was quantified by flow cytometry in the peritoneal lavage 6 h later ($n = 6$ control groups, $n = 5$ $Gch1^{fl/fl}$ Zymosan, $n = 6$ $Gch1^{fl/fl}$Tie2cre Zymosan). Lavage fluid was further analysed for total nitrite and nitrate (NOx) content by NO analyzer (**j**) and total biopterin content (**k**) ($n = 5$ $Gch1^{fl/fl}$ Control, $n = 2$ $Gch1^{fl/fl}$Tie2cre Control, $n = 5$ $Gch1^{fl/fl}$ Zymosan, $n = 6$ $Gch1^{fl/fl}$Tie2cre Zymosan, $p < 0.05$). **l** The elicited peritoneal cells were assayed for BH4 content ($n = 4$ per group, two-tailed $t$-test, $p < 0.05$). All data analysed by one-way or two-way (genotype experiments) ANOVA with Bonferroni post-test unless otherwise stated above. All 'n' represent individual biological replicates, including cell cultures from individual animals/genotype. Data shown as individual data-points with mean and SEM. *$p < 0.05$, **$p < 0.01$, ***$p < 0.001$. Source blot and flow cytometry data are available in Supplementary Figure 10 and 11

indicator dye[26]. BMDM were infected with BCG-GFP[27] and the ROS production was assessed in infected BCG-GFP+ vs uninfected BCG-GFP− cells 2 h after initial infection with BCG-GFP/IFNγ. Whilst a significant induction of ROS production was observed, this was not significantly affected by loss of BH4 (Supplementary Figure 3a). To further examine redox changes in response to infection we measured the extracellular release of $H_2O_2$, which is induced by infection and was undetectable in uninfected cells, this was significantly reduced in the $Gch1^{fl/fl}$Tie2cre cells (Supplementary Figure 3b) indicating that the changes in redox biology elicited by the loss of BH4 in macrophages is substantially more complex than has been observed in other BH4 deficient cell types[28]. Whilst in endothelial

cells ROS production is elevated as a result of eNOS uncoupling in the absence of BH4, in macrophages BH4 deficiency results in more widespread changes, but with the phagocytic oxidative burst in response to infection appearing unchanged.

**$Gch1^{fl/fl}$Tie2cre macrophages are BH4 and NO-deficient in vivo.** To assess whether the loss of NO-production had biological consequences in vivo it was important to demonstrate that the cells remained deficient in both BH4 and NO-production even in an environment where other cells were replete in BH4 levels. We confirmed Gch1 deletion rendered inflammatory cells BH4 deficient in vivo, by inducing sterile peritonitis in mice using

zymosan, to induce iNOS in recruited peritoneal exudate cells[29]. Recruitment of neutrophils and monocytes to the peritoneal cavity (Fig. 1h) was unaffected by *Gch1* deletion (Fig. 1i)[25]. Analysis of peritoneal lavage fluid in wildtype mice revealed that zymosan caused significant induction of the iNOS and tetra-hydrobiopterin pathway, measured by total biopterins and nitrite/nitrate levels (NOx), stable oxidized extracellular endproducts of nitric oxide production in vivo (Fig. 1j, k). Total biopterins (BH4, dihydrobiopterin and biopterin) rather than BH4 were assessed due to the oxidizing nature of extracellular environment causing oxidative loss of BH4 to the oxidized forms dihydrobiopterin and biopterin. In contrast, no significant elevation of biopterins or NOx levels were evident in peritoneal lavage fluid from *Gch1*[fl/fl]Tie2cre mice. Furthermore, peritoneal leukocytes from *Gch1*[fl/fl]Tie2cre animals showed barely detectable levels of intracellular BH4, in contrast to wildtype cells (Fig. 1l) indicating that *Gch1*[fl/fl]Tie2cre leukocytes remain BH4 deficient in vivo and are not rescued by systemic BH4[28].

**Gch1[fl/fl]Tie2cre mice have enhanced control of BCG infection.** Given the importance of iNOS in mycobacterial infection in mice, we next investigated the consequences of BH4 deficiency on mycobacterial infection in vivo, comparing wildtype with *Gch1*[fl/fl]Tie2cre mice. We confirmed that peripheral blood leukocyte homeostasis was not altered in *Gch1*[fl/fl]Tie2cre mice (Supplementary Figure 4a-d) and that *Gch1*[fl/fl]Tie2cre alveolar macrophages from bronchoalveolar lavage were deleted for *Gch1* (Fig. 2a), and had undetectable BH4 levels (Fig. 2b). We infected wildtype and *Gch1*[fl/fl]Tie2cre mice with BCG by intranasal instillation, and monitored the infected mice for 4 weeks. Both wildtype and *Gch1*[fl/fl]Tie2cre mice had comparable lung bacterial load, but CFU counts were significantly lower in the *Gch1*[fl/fl]Tie2cre spleens compared to wildtype mice littermates (Fig. 2c, d). BCG infection induced a significant increase in spleen:body weight ratio in *Gch1*[fl/fl]Tie2cre mice, and a non-significant increase in wildtype animals (Fig. 2e). Measurement of total biopterins in the lung and spleen homogenate showed a significant increase in infected wildtype lungs compared to un-infected controls (Fig. 2f). Total biopterin levels rather than BH4 alone were assessed due to the oxidizing nature of the homogenisation process required for BCG CFU assessment. The levels of total biopterins in the lung and spleen of infected mice were significantly lower in the *Gch1*[fl/fl]Tie2cre mice than the wildtype controls (Fig. 2f, g). The total biopterin content of BCG-infected lungs was significantly correlated with lung mycobacterial load (Fig. 2h), consistent with BH4 deficiency resulting in control of infection. BCG infection induced iNOS in the infected lung tissues from mice of both genotypes (Fig. 2i), but NOx in lung lysates was significantly lower in infected *Gch1*[fl/fl]Tie2cre mice (Fig. 2j).

**Loss of NO but not iNOS protein reduces M.tb infection.** To more stringently assess the role of macrophage nitric oxide production in a more pathogenic model we evaluated the effect of macrophage BH4 deficiency on the response to a virulent aerosolized *M.tb* challenge in *Gch1*[fl/fl]Tie2cre mice. To directly compare the effects of total iNOS deficiency with deficient iNOS-mediated NO production, we also infected *Nos2*[−/−] and control C57BL/6J mice under the same conditions. Inclusion of this control group was of particular importance given the variability in onset of susceptibility seen in previous studies[5–7]. As expected, *Nos2*[−/−] mice developed dramatic macroscopic lung and spleen pathology after *M.tb* infection, with numerous large lung granulomatous lesions and splenomegaly (Fig. 3a and Supplementary Figure 5). In contrast, none of these features were prominent in

either the *Gch1*[fl/fl]Tie2cre or the wildtype animals. Both wildtype control mice and *Gch1*[fl/fl]Tie2cre mice gained weight over 4 weeks after infection (Fig. 3b). In contrast, *Nos2*[−/−] mice showed a steady decline in weight from 2 weeks post-infection, and required termination of the experiment 23 days post-infection at a pre-specified humane endpoint. The *Nos2*[−/−] mice had significantly higher lung and spleen CFU counts than the C57BL/6J control group (Fig. 3c, d), whereas CFU counts in the *Gch1*[fl/fl]Tie2cre mice were not different from their wildtype littermate controls. Body weight correlated with lung CFU counts in individual animals (Fig. 3e), with *Gch1*[fl/fl]Tie2cre mice having highest bodyweights, in keeping with the lowest lung CFU. Immunohistochemistry of lung granuloma confirmed iNOS expression and macrophage infiltration within the granulomata of wildtype and *Gch1*[fl/fl]Tie2cre mice that was absent in *Nos2*[−/−] mice (Fig. 3a and Supplementary Figure 6).

To further assess the control of *M.tb* infection in *Gch1*[fl/fl]Tie2cre mice, we undertook more prolonged infection experiments. Six weeks after infection with aerosolized *M.tb*, *Gch1*[fl/fl]Tie2cre mice had significantly lower CFU per lung than their wildtype littermate controls (Fig. 3f). Spleen CFU counts were comparable between groups (Fig. 3g). Whilst the reduced extrapulmonary spread of infection, seen when mice were infected with BCG, was not maintained when *Gch1*[fl/fl]Tie2cre mice are exposed to the more virulent *M.tb* infection, this stringent test of susceptibility shows a clear reduced susceptibility in the lung at a later 6-week timepoint. As expected, *Gch1* gene expression in the infected *Gch1*[fl/fl]Tie2cre tissues (Supplementary Figure 7) was significantly lower whereas expression of other key inflammatory genes, *Nos2*, *Tnfα* and *Cd68*, were unaffected, indicating preservation of the inflammatory response.

**Gene expression reveals signature unique to loss of Gch1.** Due to the multiple changes in the redox biology of the BH4 deficient macrophages we next sought to use gene expression analysis to gain insight into the mechanism of the enhanced control of mycobacterial infection observed in *Gch1*-deleted, BH4-deficient macrophages. We performed a mouse whole genome array analysis on *Gch1*[fl/fl]Tie2cre and *Nos2*[−/−] macrophages, infected with BCG in the presence of IFNγ for 24 h (to induce robust iNOS expression), compared with control uninfected macrophages for each genotype. This experimental design enables us to specifically compare the conditions of loss of iNOS and loss of iNOS-mediated NO-production in the context of BH4 deficiency.

We identified differentially expressed genes in response to BCG/IFNγ in wildtype mice. As expected, BCG/IFNγ caused major changes in macrophage gene expression profiles, with 12,412 probes being differentially expressed (Supplementary Data 1). Heatmap analysis of the 1000 most regulated genes showed a consistent activation across all genotypes (Supplementary Figure 8). Pathway analysis of the most significantly regulated genes, using the Ingenuity Pathway Analysis (IPA) tool, indicated activation of pathways associated with inflammation and infection (Supplementary Figure 8 and Supplementary Data 2) as expected.

We next compared the effect of loss of *Gch1* and *Nos2* in both BCG/IFNγ treated and uninfected macrophages. We observed only minor variation in uninfected macrophages between *Gch1*[fl/fl]Tie2cre and *Nos2*[−/−] macrophages and their controls, with only 77 and 26 probes, respectively, being differentially expressed (Fig. 4a and Supplementary Data 3). In contrast, when infected with BCG, 779 (*Gch1*[fl/fl]Tie2cre vs *Gch1*[fl/fl]) and 172 (*Nos2*[−/−] vs *Nos2*[+/+]) probes showed differential expression (Fig. 4a and Supplementary Data 3), with 737 probes being unique to the gene

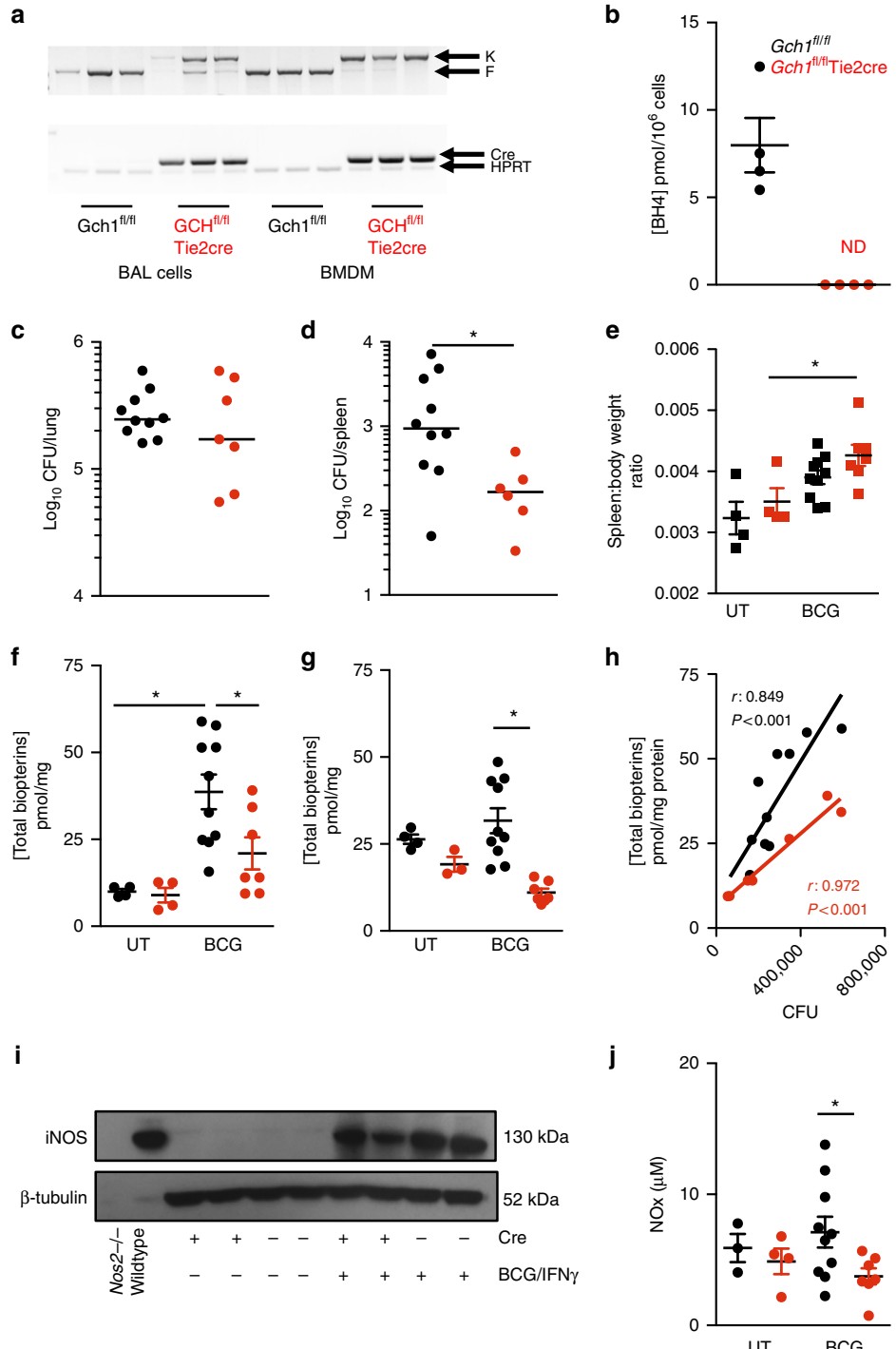

list for infected $Gch1^{fl/fl}$Tie2cre macrophages. IPA pathway analysis revealed significant activation, in $Gch1^{fl/fl}$Tie2cre macrophages, of key functions relating to cell–cell signaling, inflammatory response and lipid metabolism (Fig. 4b, c). In particular, $Gch1$ deletion up-regulated pathways regulating myeloid cell recruitment, neutrophil activation, phagocyte degranulation, and sterol synthesis, whereas cell death and apoptosis of leukocytes were significantly down-regulated (Fig. 4b, c, Supplementary Data 4). Analysis of potential upstream regulators identified enhanced signaling via TLRs 2/4/5/9 and inflammatory cytokine/cytokine receptors including IL1, TNF superfamily, and CCL5 (Fig. 4d, Supplementary Data 5). These data indicated that part of

the inflammatory response to infection appeared to be enhanced in the BH4 deficient macrophages. We validated the presence of genes that show regulation only as a result of BH4 deficiency ($As3mt$) and as a result of loss of both BH4 and iNOS ($Slc40a1$) in our gene array study in an independent cohort of cells (Fig. 5). These data confirmed the gene expression changes observed in our earlier study of LPS/IFNγ stimulated macrophages, with the identification of similar gene expression changes in both $As3mt$ and $Slc40a1$[25] and a similar prediction of decreased NRF2-mediated gene expression (Fig. 4d). However, it also demonstrated striking divergence regulation of key genes such as IL-6 which showed decreased activation following LPS/IFNγ treatment

**Fig. 2** Absence of *Gch1* improves the resistance of mice to in vivo BCG infection. **a** The excision of the floxed *Gch1* allele in both cells from WT and *Gch1fl/fl*Tie2cre recovered by bronchial-alveolar lavage (BAL) was confirmed by genomic PCR, with analysis of BMDM used as a positive control for cre activity: knockout allele (K) floxed allele (F). The presence or absence of the cre transgene was confirmed using HPRT as a housekeeping gene. **b** The BH4 (pmol/$10^6$cells) content of the BAL cells was analysed by HPLC ($n = 4$ per genotype, two-tailed *t*-test, $p < 0.05$). **c**, **d** *Gch1fl/fl*Tie2cre (red) and wildtype (black) mice were infected intranasally with $5 \times 10^6$ colony forming units (CFU) BCG Pasteur. Four weeks following infection the lungs and spleen were removed and homogenised for analysis along with tissues from a matched cohort of uninfected animals. The bacterial load of the organs was enumerated by CFU count from lung (**c**) and spleen (**d**). Each dot represents one animal, line represents median CFU of each group ($n = 10$ *Gch1fl/fl*, $n = 6$ *Gch1fl/fl*Tie2cre (Spleen), $n = 7$ *Gch1fl/fl*Tie2cre (Lung) per group), $p < 0.01$ (Mann–Whitney test). **e** Spleen to body weight ratio was also calculated. Remaining tissue homogenate was used to measure total biopterin content, in lungs (**f**) and spleen (**g**) ($n = 10$ *Gch1fl/fl*, $n = 7$ *Gch1fl/fl*Tie2cre Spleen (infected) and $n = 4$/genotype (uninfected), $p < 0.05$). Total lung biopterin content was correlated to lung CFU in all animals. Pearson r $p < 0.05$ (**h**). **i** iNOS protein expression was confirmed by western blotting, with ß-tubulin loading control ($n = 2$ per group shown, representative of $n = 3$ independent biological replicates assayed in total). An NO analyzer was used to measure NOx content in the lung homogenates ($p < 0.05$) (**j**). All data analsyed with ANOVA with Bonferroni post-test (apart from (**b-d**) and (**h**) as above). Data shown as individual data-points with mean and SEM (apart from (**c**) and (**d**) as above). *$p < 0.05$. Source gel and blot images are available in Supplementary Figure 10

but enhanced expression following BCG/IFNγ treatment (Fig. 4c), indicating exposure to a live pathogen causes a unique pattern of gene regulation that cannot be adequately modelled by the use of other proinflammatory stimuli.

To more broadly evaluate the entire dataset, we performed unsupervised Gene Set Enrichment Analysis (GSEA) to identify coordinated regulation of annotated groups of genes, implicated in defined cellular processes[30]. We performed Preranked GSEA using the GO term-defined gene sets relating to biological processes (MSigDB C5 BP) followed by network analysis of the most significantly altered gene sets using Enrichment Map[30] (Fig. 6a, b, Supplementary Data 6). We then compared the lists of significantly altered gene sets for each genotype comparison to identify which were shared between genotypes and which were unique to the loss of *Gch1* (Supplementary Figure 9a) these new gene set sub groups were again visualized using Enrichment Map (Fig. 6c and Supplementary Figure 9B). The majority of gene sets relating to DNA repair, protein catabolism, vesicular trafficking and ribosome activity were found to be common to the loss of either *Gch1* or *Nos2* (Supplementary Figure 9b). However, a number of gene signatures were unique to the loss of *Gch1* (Fig. 6c, d). These gene sets were related to fundamental cellular processes such as metabolism and cellular detoxification as well as to processes related to inflammation and infection such as lysosomal function, apoptosis and cell senescence and innate immune. To confirm the biological relevance and validity of these findings, we performed a second round of GSEA using gene sets relating to canonical pathways and curated datasets (MSigDB C2 CP). This analysis again identified a number of gene sets that were either shared by both genotypes or unique to the loss of *Gch1* (Fig. 7 and Supplementary Data 6). *Gch1* unique gene sets again included those related to lysosome function, inflammation, DNA damage response (including p53 pathway) and lipid and glucose metabolism confirming our findings.

*Gch1fl/fl*Tie2cre macrophages showed reduced IL-6 gene and protein expression in response to BCG/IFNγ, as predicted by the pathway analysis (Fig. 8). Under the conditions used for gene expression analysis no alterations in cell viability were seen (Fig. 9a). Under conditions of enhanced greater stress, with cells being infected with BCG/IFNγ (MOI 5:1) we observed a decreased macrophage cell survival, associated with increased apoptosis of infected cells, in the absence of *Gch1* again confirming that the gene expression analysis is predictive of alterations in macrophage cell biology. Gene sets related to the Sterol/Cholesterol pathway can have multiple effects on cell lipid handling including altered cholesterol handling and lipid droplet format, altered oxysterol production and alterations to LXR/RXR ligands. We tested the ability of BH4 deficient cells to form lipid droplets in response to BCG/IFNγ, using a recently reported

method and droplet formation still occurred in the absence of NO (Fig. 9b)[31].

**Gch1 expression alters mycobacterial growth in vitro**. To determine whether *Gch1* and BH4 directly alter control of mycobacterial infection in macrophages, we infected BMDM macrophages in vitro with BCG and quantified the control of mycobacterial infection using the mycobacterial growth inhibition assay (MGIA)[32]. After BCG infection, CFU counts were significantly higher in *Nos2*$^{-/-}$ macrophages than their wildtype controls (Fig. 10a). In contrast, *Gch1fl/fl*Tie2cre macrophages had significantly lower CFU counts than their wildtype controls (Fig. 10b). This indicates that BH4-deficient macrophages alone have an enhanced cell-intrinsic mechanism for control of mycobacterial growth.

Having shown that *Gch1* deletion increases the control mycobacterial infection by murine macrophages, we next tested whether a similar association between *GCH1* expression and mycobacterial control is seen in human cells. PBMCs from BCG-vaccinated infants, from a previously reported vaccine efficacy study[33], were used to perform an in vitro MGIA to assess control of BCG infection. In parallel, samples were infected with BCG overnight and underwent analysis for *GCH1* and *NOS2* gene expression. There was a significant positive correlation between *GCH1* gene expression and increased mycobacterial CFU in the direct PBMC MGIA assay (Fig. 10c). In contrast, there was no correlation between *NOS2* gene expression and mycobacterial control (Fig. 10d). Thus, lower *GCH1* expression in human PBMCs is also associated with increased control of mycobacterial infection, independent of PBMC *NOS2* expression.

## Discussion

Our findings suggest that re-evaluation of the role, and mechanism of action, of NO in macrophage anti-mycobacterial activity is required. Along with other recent studies our data show that a direct cytotoxic role for NO within the macrophage is not the only, and possibly not the main, function for NO[34,35]. Whilst direct exposure of mycobacteria to nitric oxide gas has been confirmed to mediate mycobacterial killing[10], our observations suggest that NO production by leukocyte-expressed iNOS is not the only mechanism by which iNOS can control *M.tb* and indeed a lack of NO maybe beneficial in some circumstances. A rein-terpretation of the role of NO in TB has important implications for understanding the apparent discordance between the strong evidence for iNOS and NO in the control of mycobacteria in murine models, compared with a less clearly defined role for iNOS and NO system in human disease[18–20]. Although iNOS is expressed in inflammatory conditions such as TB[21,22] the ability

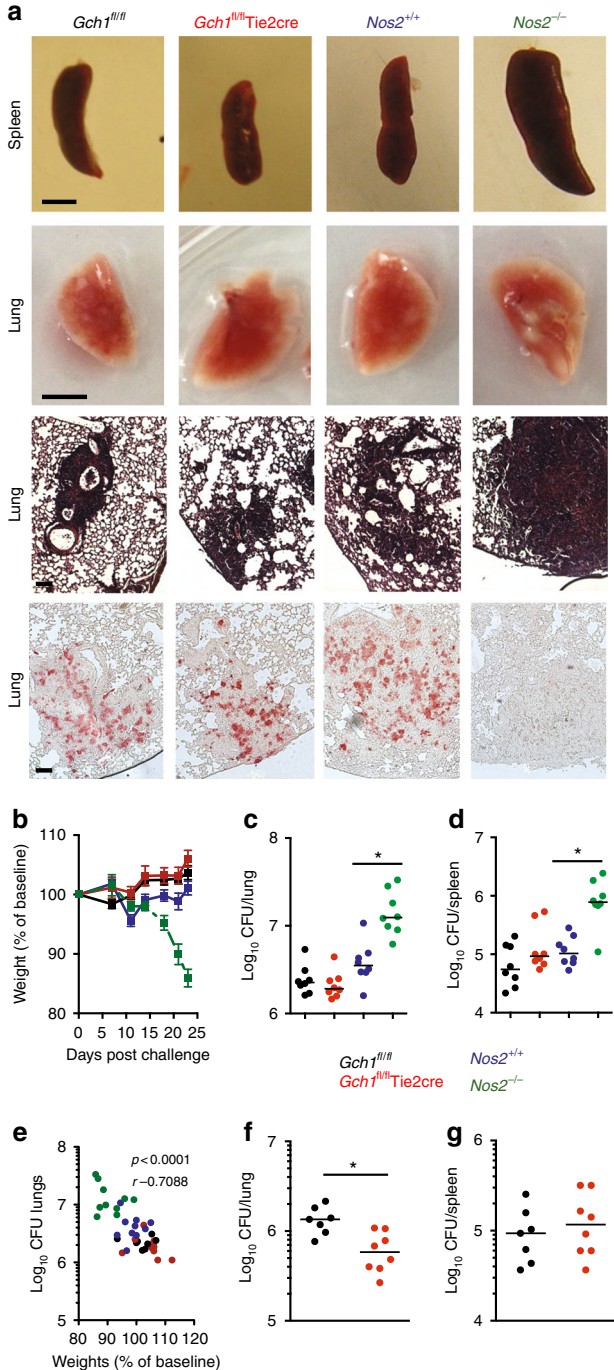

**Fig. 3** Absence of *Gch1* increases resistance of mice to virulent *M.tb* infection. *Gch1*<sup>fl/fl</sup>Tie2cre (red) and wildtype (black), *Nos2*<sup>−/−</sup> (green) and *Nos2*<sup>+/+</sup> (blue) mice were infected with 150 CFU *M.tb* (Erdman strain) by aerosol inhalation. 23 days following infection the lungs and spleen were removed for analysis. Representative organs were imaged from each genotype prior to *n* = 2 tissues per genotype being embedded for histochemical analysis. Sections were cut from all tissues and stained with H&E or subjected to anti-iNOS immunohistochemistry (**a**). Animals were weighed at regular intervals throughout the experiment and the mean weight change and SEM per group was plotted (**b**). Lungs and spleen were homogenised and the mycobacterial load of the organs was enumerated by CFU count from lung (**c**) and spleen (**d**) (*n* = 8 per group, * *p* < 0.01) (Mann–Whitney). Each dot represents one mouse, line in each group represents the median CFU. **e** Weights negatively correlated with CFU counts. Pearson *r p* < 0.05. **f** *Gch1*<sup>fl/fl</sup>Tie2cre (red) and *Gch1*<sup>fl/fl</sup> (black) mice were infected as before but this time lungs and spleen were removed 6 weeks following infection for analysis. Lungs and spleen were homogenised and the mycobacterial load of the organs was enumerated by CFU count from lung (**f**) and spleen (**g**) *n* = 7 *Gch1*<sup>fl/fl</sup>, *n* = 8 *Gch1*<sup>fl/fl</sup>Tie2cre, *p* < 0.01 (Mann–Whitney). The finding in (**f**) was replicated on two occasions in independent experiments. Scale bars: upper panels 0.5 cm, lower panels 100 um

neopterin, that has been used as a biomarker in TB[37]. The *GCH1* gene has been identified as a marker of active disease in TB patients in several gene expression studies[38,39]. However, these studies did not report any further association other than increased expression in infected individuals, compared to healthy or disease control patients. In contrast, we now demonstrate that *GCH1* expression, rather than *NOS2* expression, correlates with control of mycobacterial growth in human PBMC. Indeed, a previous study has reported increased mycobacterial growth in infected human macrophages in the presence of exogenous BH4 supplementation[40]. Thus, human macrophages may share the NO-independent means of controlling mycobacteria under conditions of low BH4 availability, suggesting that inhibition of macrophage BH4 synthesis may provide a therapeutic target for TB.

INOS is increasingly recognized as having a signaling role in TB by preventing excessive inflammation and tissue damage. A recent study identified HIF-1α, downstream of iNOS activation, as a crucial mediator of IFNγ mediated response to *M.tb* infection[35,41]. *Nos2*<sup>−/−</sup> mice were used to demonstrate that iNOS was a crucial stabilizer of HIF-1α, resulting in increased HIF-1α signaling[41]. Genetic deletion of either iNOS or HIF-1α resulted in reduced control of mycobacteria by macrophages. Critically iNOS and HIF-1α mediated opposing roles in the regulation of a panel of pro-inflammatory cytokines, including IL-6, IL-1 and neutrophil chemokines, indicating iNOS has wider roles in the cell signaling than via HIF-1α[41]. The ability of NO to chemically modify proteins through nitrosylation of cysteine residues is one way in which iNOS can transduce signals within the cell. In the case of mycobacterial infection, nitrosylation of NLRP3 prevents inflammasome activation thus reducing the production of important pro-inflammatory mediators such as IL-1[42]. Accordingly IL-1 and lipoxygenase mediated pathways were shown to be overactive in *Nos2*<sup>−/−</sup> mice with the resultant excessive neutrophil recruitment and tissue damage supporting enhanced mycobacterial growth[34]. BH4 deficient NO-deficient macrophages, appear to show transcriptional changes in similar gene sets and molecules highlighted in earlier studies in *Nos2*<sup>−/−</sup> cells and tissues[34,41]. Intriguingly, cellular pathways identified through analysis of iNOS protein–protein interactions[17], namely ribosomal and proteosomal pathways, are found amongst the shared

of human macrophages to produce nitric oxide is controversial[36]. SNPs in the *NOS2* gene have been associated with altered susceptibility to disease, with many identified SNPs sitting in the promoter region and associated with increased gene transcription and associated with protection from TB[3]. These data do indicate that alterations in iNOS biology have an impact on human susceptibility, however our data would suggest that this may not be a simple association with NO cytotoxicity, but that the relative expression of *NOS2* and *GCH1* is likely to be key.

iNOS is expressed in macrophages from TB patients, but human macrophages exhibit exon skipping during transcription of 6-pyruvoyl tetrahydropterin synthase (PTPS), the second step in BH4 production, which leads to low-level production on BH4 in these cells, and accumulation of the proximal metabolite,

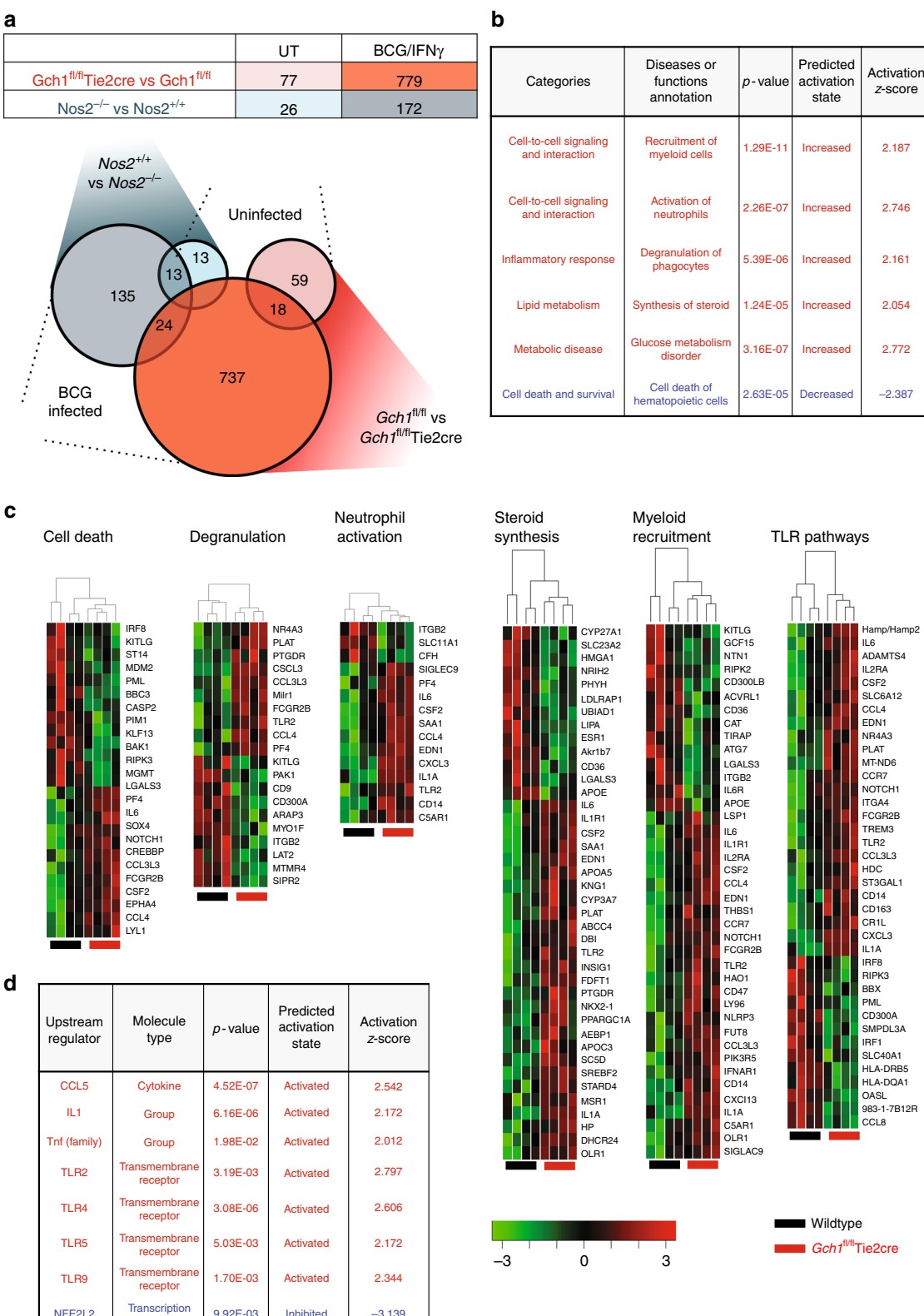

gene sets found in both BH4-deficient and iNOS-deficient macrophages, hinting that NO production by iNOS may also modulate its interaction with other proteins. The endothelin system is a key effector of NOS enzymes in other cell types and also showed alterations in gene expression levels in the absence of BH4, but expression of both protective (ET-1) and non-protective (ETB) elements of the system were downregulated[43]. So whilst some

signaling functions attributed to iNOS are linked to nitric oxide, the potential contribution of other signaling pathways is overlooked. The critical difference between our study and previously published analyses of the Nos2[-/-] mouse is that selective macrophage BH4 deficiency results in loss of NO production, rather than loss of all potential iNOS functions, and this results in increased control of mycobacterial infection.

**Fig. 4** Absence of *Gch1* alters macrophage gene expression in response to infection. Macrophages were infected with BCG (MOI 1:1)/IFNγ (10 ng/ml) for 24 h or incubated for 24 h in the absence of infection ($n = 4$ per genotype). RNA was extracted and a whole mouse genome gene expression array was performed. **a** Genes were determined as significantly regulated by genotype by comparison to their relevant wildtype control with $p < 0.05$ (adjusted for multiple testing). The number of genes significantly regulated by genotype for each condition was calculated. The number of genes shared between genotypes and conditions is shown by Venn Diagram. **b** Selected relevant gene functional annotations determined with significantly increased or decreased expression in BCG infected wildtype vs *Gch1*<sup>fl/fl</sup>Tie2cre macrophages, as determined using IPA analysis with $Z$ score $> 2$ and a *p*-value for overlap $p < 0.01$. **c** Heatmaps for genes within individual gene functional annotations determined in (**c**, **d**). **d** Selected relevant predicted upstream regulators with significantly increased or decreased activation in BCG infected wildtype vs *Gch1*<sup>fl/fl</sup>Tie2cre macrophages, as determined using IPA analysis with $Z$ score $> 2$ and a *p*-value for overlap $p < 0.01$

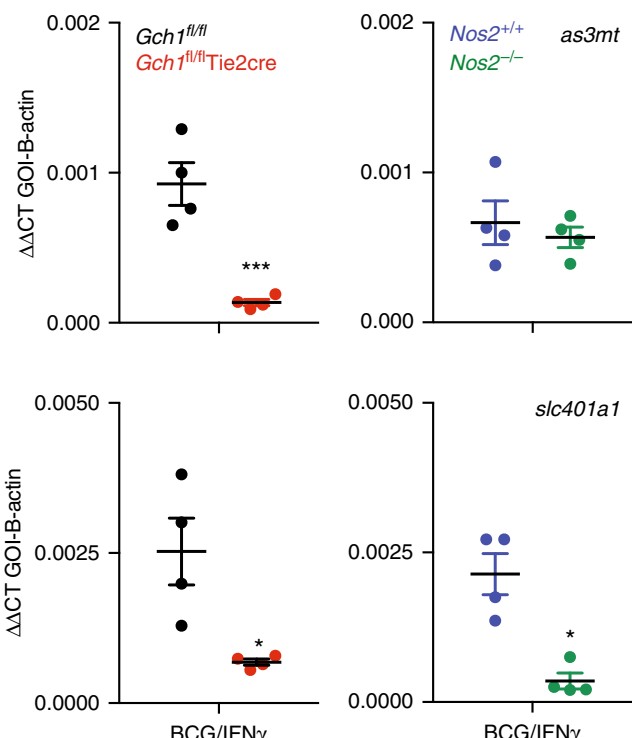

**Fig. 5** Divergent gene expression changes in BH4 and iNOS-deficient macrophages. BMDM were prepared from *Gch1*<sup>fl/fl</sup>, *Gch1*<sup>fl/fl</sup>Tie2cre, *Nos2*<sup>+/+</sup> and *Nos2*<sup>−/−</sup> age-matched animals. Macrophages were infected at a MOI of 1:1 with BCG in the presence of 10 ng/ml IFNγ, or left uninfected followed by RNA extraction. Gene expression was quantified using TaqMan gene expression assays. GOI Gene of Interest. Data was analysed by two-tailed *T*-test * $p < 0.05$, error bars ± SEM

The loss of macrophage BH4 is sufficient to enhance control of mycobacterial infection by the cell itself. Upon infection with mycobacteria, mycobacterial killing is assumed to occur by the maturation of the uptake phagosome and fusion with the lysosome to form the phagolysosome[44]. While the cytotoxic capacity of the cell was assumed to then reside in the ability to produce RNI and ROS species, increasingly evidence suggests this is not the case. The role of ROS in direct mycobacterial control is as controversial as the role of nitric oxide. The loss of ROS production in *p47*<sup>PHOX−/−</sup> mice resulted in only a transient decrease in bacterial control, and human chronic granulomatous disease patients, who lack myeloid cell ROS production, do not show increased susceptibility to TB[45,46]. An interpretation of our findings is that *Nos2*<sup>−/−</sup> cells lack all aspects of iNOS activity (both NO and ROS-mediated), whereas *Gch1*<sup>fl/fl</sup>Tie2cre cells lack NO production but have preserved ROS production, that may mediate mycobacterial killing. A direct effect of ROS on mycobacterial killing seems less likely, given both the complicated pattern of altered ROS production in the absence of BH4 and

equivocal evidence for mycobacterial killing by ROS. Nevertheless, altered ROS production may exert important effects through altered downstream redox signaling. Increasingly the fate of lysosomal bacteria through the engagement of autophagy and intracellular degradation machinery is highlighted as playing a role in bacterial killing[44,47,48]. Similarly macrophage cell death by apoptosis is associated with decreased viability of mycobacteria[49,50]. The extent to which ROS, nitric oxide and iNOS act as signaling molecules in coordination of these responses may link older and newer findings. The altered lysosomal function and cell survival signals we identified in *Gch1* deficient macrophages could clearly have implications for mycobacterial control. Additionally, a BH4 specific alteration in metabolic and mitochondrial gene sets in infected macrophages, has similar potential as *M.tb* causes a profound shift in cellular metabolism towards aerobic glycolysis, which was evident in leukocytes within the lung[51]. Control of mycobacterial infection downstream of IFNγ and iNOS was shown to involve activation of aerobic glycolysis, which amplified macrophage activation[35]. Induction of aerobic glycolysis by human macrophages following mycobacterial infection is reported to facilitate control of bacterial replication, with inhibition of this process associated with enhanced mycobacterial survival[52]. As extracellular $H_2O_2$ accumulation has been linked with activity of the mitochondrial electron transport chain the altered $H_2O_2$ accumulation seen following infection in this study may indicate metabolic changes in the absence of macrophage BH4[53].

We have used a genetic approach to identify new mechanisms linking macrophage iNOS and NO to the control of mycobacterial infection in vitro and in vivo, using cell-specific BH4 deficiency, achieved by conditional *Gch1* deletion. NOS enzymes can function independently of NO both through protein–protein interactions and in an 'uncoupled' state in the absence of BH4, whereby L-arginine is not oxidized to NO and superoxide is produced by the reductase domain of the enzyme[14,15,17]. Thus, equating NOS activity with nitric oxide production alone potentially overlooks important alternative signaling pathways. Whilst these pathways are dominated by superoxide production from NOS in other BH4-deficient NOS expressing cells, such as endothelial cells[28], the redox biology of macrophages lacking BH4 appears more complex. It has previously been reported that BH4 has important effects on cellular metabolism independent of its effect on NOS biology[54]. Knockdown of *Gch1* in the absence of any NOS expression resulted in alterations in bioenergetic metabolism[54]. So whilst the transcriptional effects seen on macrophage metabolism may be important, these may not all be dependent iNOS expression. Tetrahydrobiopterin has other NOS-independent functions as it is also a co-factor for other enzymes such as alkylglycerol monooxygenase (AGMO), which modulates cellular lipid signaling, and catecholamine-related enzymes[15,55]. These NOS-independent roles for BH4 in macrophage biology are still of particular relevance to humans due to the low levels of BH4 present in human macrophages, such that alterations in cell function relating to both reduced NO-production from iNOS and

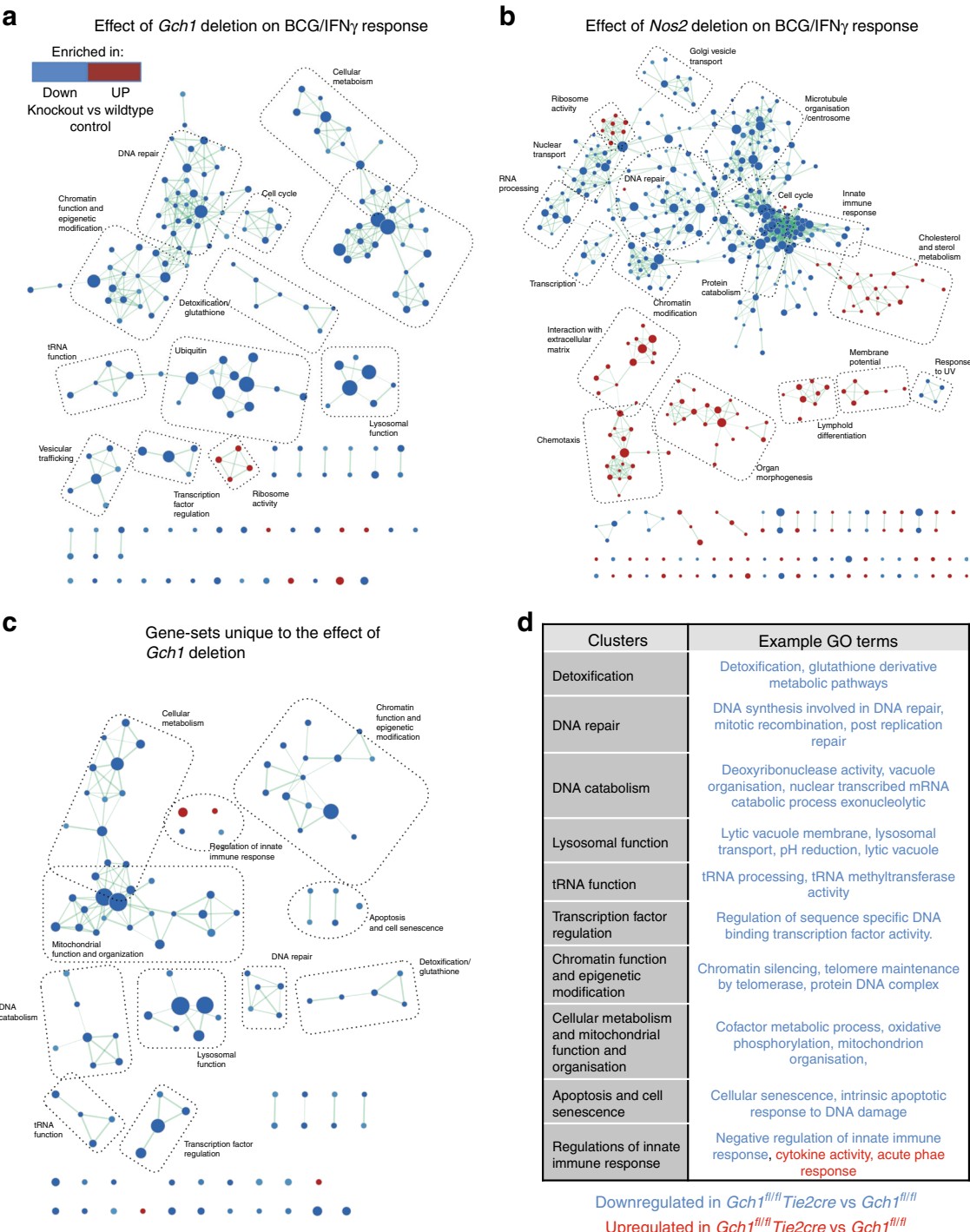

**Fig. 6** GSEA identifies unique gene expression patterns in *Gch1* cells. Gene expression data from both *Gch1* and *Nos2* deficient macrophages were analysed for GO term (biological processes) enrichment by Preranked gene-set enrichment analysis (GSEA) vs their wildtype control. The result was visualized as a network of gene sets (nodes) connected by their similarity (edges), compiled in Cytoscape using the Enrichment Map plug in. Node size represents the gene-set size and edge thickness represents the degree of overlap between neighboring gene sets. The GSEA output files were given analysed with a statistical cutoff set at *p* < 0.005 and FDR *q* < 0.1. The individual Enrichment Maps were plotted in Cytoscape for infected **a** *Gch1* and **b** *Nos2* macrophages and clusters were annotated by hand. To determine if regulated genesets were shared between the *Gch1* and *Nos2* knockout macrophages the significantly regulated gene sets were compared and a subset of genesets unique to *Gch1* deficient macrophages was created, this subset (**c**) was plotted using Enrichment Map and the clusters annotated. **d** A table summarizing the GO-term clusters, with exemplar GO terms

iNOS-independent BH4-dependent changes in macrophage biology have key translational relevance.

We demonstrate an unanticipated aspect of BH4/iNOS biology that mediates enhanced control of mycobacterial infection through mechanisms that are independent of iNOS-mediated NO production. *Gch1* and BH4 exert striking effects on the control of mycobacterial infection in macrophages independent of NO-production, in both mice and humans. The BH4-deficient

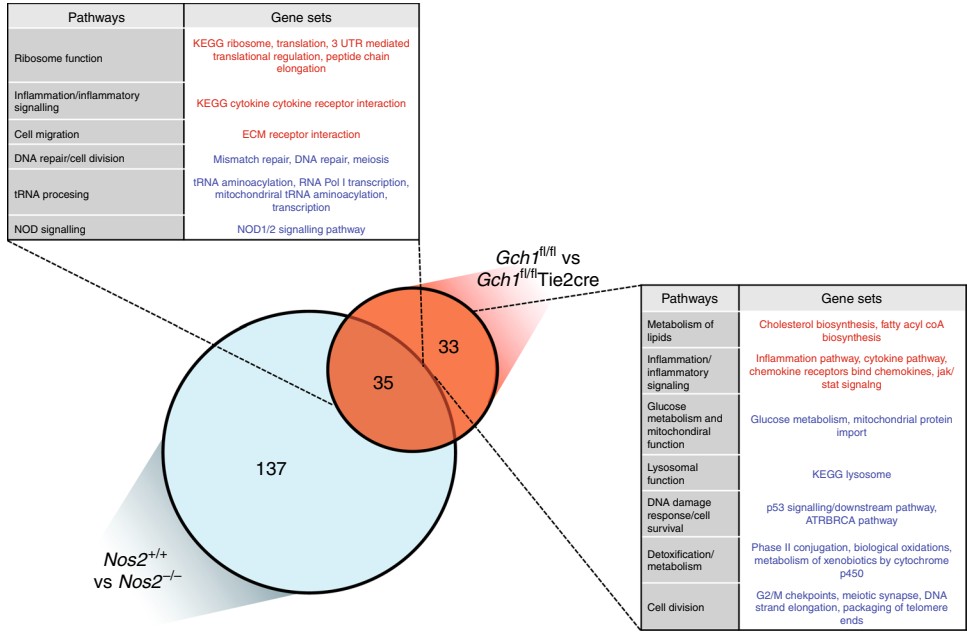

**Fig. 7** GSEA using curated pathways identifies shared and unique genesets. Enriched gene sets in BCG infected macrophages, identified by Preranked GSEA (MSigDb, C2 CP annotations set). To determine how far regulated geneset were shared between the *Gch1* and *Nos2* knockout macrophages or unique to the loss of *Gch1* the gene set lists passing $p < 0.005$, FDR $q < 0.1$ in both analyses were compared and key genesets were are highlighted in the tables

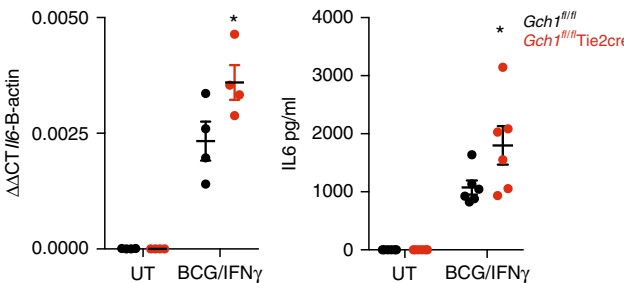

**Fig. 8** Gene expression changes result in functional changes is cytokine expression. Macrophages were infected with BCG (MOI 1:1)/IFNγ (10 ng/ml) for 24 h or incubated for 24 h in the absence of infection followed by RNA extraction or ELISA analysis of the supernatant. IL-6 gene expression was quantified using TaqMan gene expression assays. Data was analysed by two-tailed *t*-test, * $p < 0.05$, error bars ± SEM

*Gch1*fl/fl Tie2cre mouse may provide a better in vivo model to investigate novel human therapeutics and vaccinations than either wildtype mice, or *Nos2*−/− mice, due to the strong parallels with human macrophage biology. Exploiting this model could provide novel targets for HDT to control mycobacterial infection that are not possible to study in traditional models.

## Methods
**Infant PBMC samples.** The correlates of risk samples used in this study formed part of larger cohort from a phase IIb efficacy trial (ClinicalTrials.gov number NCT00953927)[33]. The trial was approved by the University of Cape Town, Faculty of Health Sciences, Human Research Ethics Committee, Oxford University Tropical Research Ethics Committee and the Medicines Control Council of South Africa. Informed consent was obtained from the mothers of all infants. Briefly, 2797 healthy infants (4–6 months) who had received BCG vaccination within 7 days of birth were randomly divided into either receiving MVA85A ($n = 1399$) or Candida as placebo ($n = 1398$). Infants were monitored for up to 37 months. The primary outcome of this study was safety but the efficacy of MVA85A vaccine was also assessed, with primary study endpoint the incidence of tuberculosis. MVA85A did not significantly improve BCG efficacy (17.3%, 95% CI −31.9 to 48.2). Some of these samples were used in an identification of correlates of risk study ($n = 258$)[56].

A number of assays (on a priority list) were performed on whole blood and PBMC samples from day of enrolment (D0) and 28 days post vaccination (D28). The MGIA assay was only performed if enough cells were available. For the analysis in this manuscript, PBMC samples ($n = 78$) from D0 were used to avoid confounding effects of MVA85A vaccination.

**Animal studies.** All animal studies were conducted with ethical approval from the Local Ethical Review Committee at the University of Oxford and in accordance with the UK Home Office regulations (Guidance on the Operation of Animals, Scientific Procedures Act, 1986). Mice were housed in ventilated cages with a 12-h light/dark cycle and controlled temperature (20–22 °C), and fed normal chow and water ad libitum. For in vivo experiments adult female mice >10 weeks of age were used for all studies, whole litters were randomly assigned to experimental groups, with heterozygous breeding strategies providing a mix of genotypes within each litter. Experiments using littermate animals were performed blinded to the genotypes of the individual animals. For homozygous strains in vivo experiments were blinded to the genotype by labeling of the cages by a group code, rather than genotype. Mice had received no prior procedures (including acting as breeding stock) prior to use in experiments for this manuscript.

***Gch1*fl/fl Tie2cre and *Gch1*fl/fl mice.** A novel mouse model of BH4 deficiency was created by conditional deletion of *Gch1*, that encodes GTP cyclohydrolase I the rate-limiting enzyme in BH4 biosynthesis. Exons 2 and 3 of *Gch1*, encoding for the active site of GTPCH protein, were flanked by loxP sites in a targeting construct that was used to produce *Gch1*fl/fl mice following homologous recombination in ES cells. Mice have been backcrossed for >8 generations to C57BL/6J. These mice were crossed with Tie2cre transgenic mice to produce *Gch1*fl/fl Tie2cre mice where *Gch1* is deleted in endothelial and bone marrow-derived cells[28]. The Tie2cre transgene is active in the female germline, as such only male Tie2cre heterozygote animals are used to establish breeding pairs to maintain conditional expression. Mice were genotyped using primers targeted against the floxed *Gch1* allele, and in a separate reaction for the presence of the cre sequence. Experiments were performed using bone marrow isolated from adult (>10-week old) female littermate animals. Animals were genotyped using DNA prepared from ear notches using the following PCR primers:
Cre:
5′ GCATAACCAGTGAAACAGCATTGCTG 3′
5′ GGACATGTTCAGGGATCGCCAGGCG 3′
*Gch1* floxed allele:
5′ GTCCTTGGTCTCAGTAAACTTGCCAGG 3′
5′ GCCCAGCCAAGGATAGATGCAG 3′

***Nos2*−/− mice.** *Nos2*−/− mice and C57BL/6J control mice were obtained from Jackson Labs (Bar Harbor, US) and Charles River (UK)[57]. Experiments were

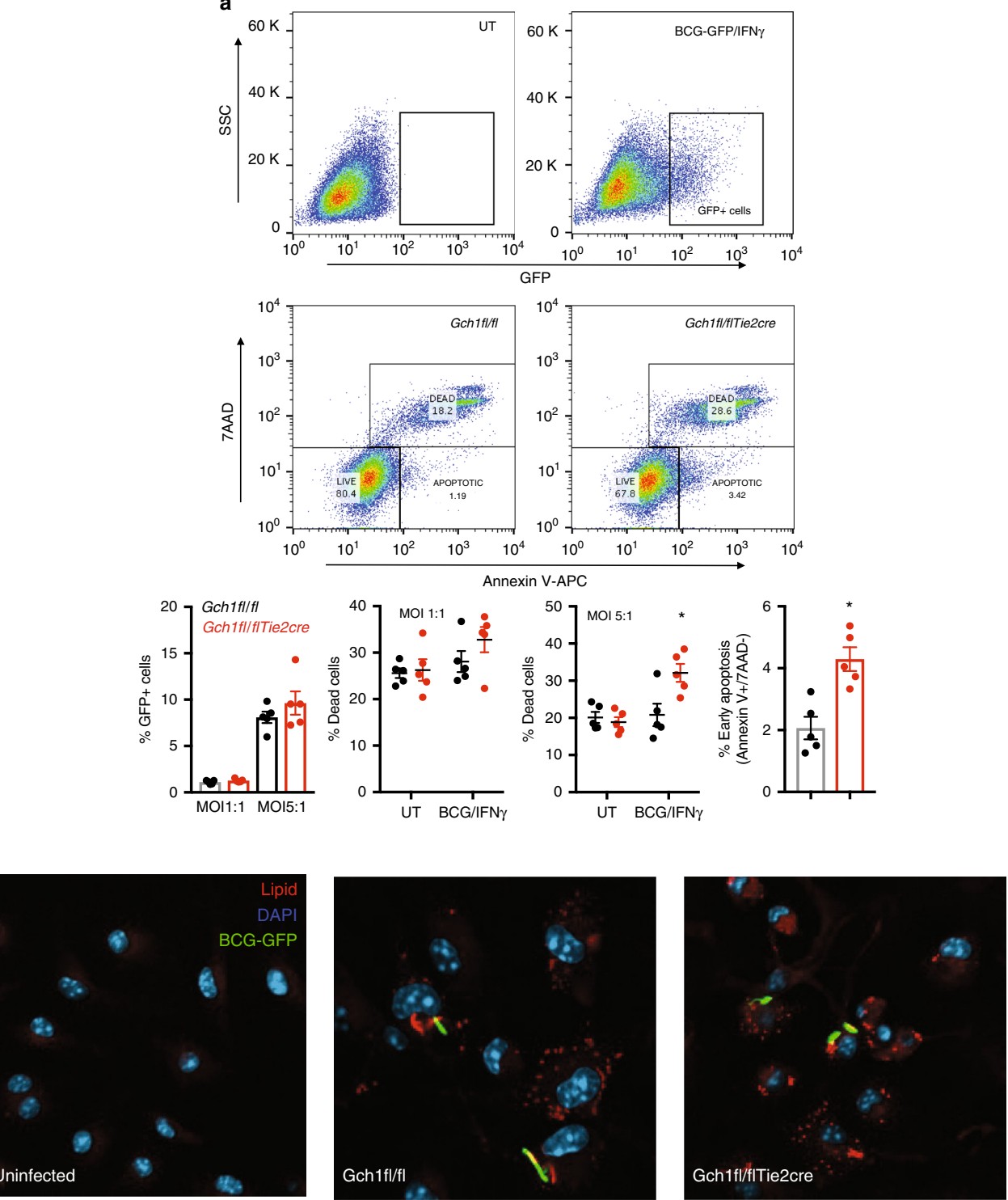

**Fig. 9** BH4 deficiency alters infected cell viability. **a** *Gch1*fl/flTie2cre and *Gch1*fl/fl macrophages were stimulated IFNγ (10 ng/ml) and BCG (MOI 1:1 or 5:1) or left untreated overnight. At the end of the assay the cells were harvested and incubated in buffer containing 7AAD and Annexin V-APC for 15 min at room temperature. Staining was terminated by addition of buffer and cells were analysed by flow cytometry immediately. Cell death was defined as Annexin V+/7AAD+. Infection was confirmed by detection of GFP and early apoptosis (Annexin V+/7AAD−) was assessed in BCG-GFP+ cells infected at MOI 5:1. Data was analysed by 2-way ANOVA *$p < 0.05$ with Bonferroni post-test or two-tailed t-test ($n = 5$/genotype). **b** *Gch1*fl/flTie2cre and *Gch1*fl/fl macrophages were stimulated with IFNγ (10 ng/ml) and BCG (MOI 1:1) or left untreated overnight in the presence of 2% serum as a lipid source. At the end of the assay the cells were fixed and neutral lipids were stained using LipidTox dye and nuclei stained with DAPI. Images are representative of 4 individual macrophage cultures. Error bars ± SEM

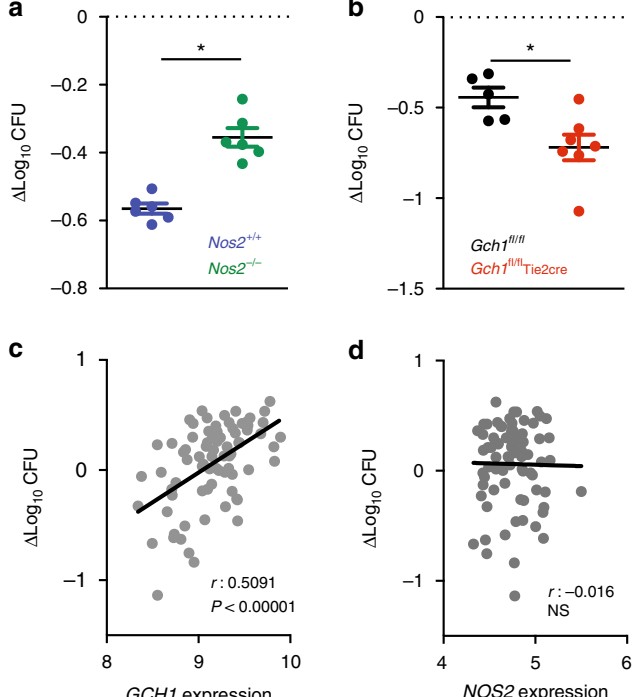

**Fig. 10** *Gch1* expression inversely correlates with control of BCG infection in vitro. **a**, **b** A mycobacterial growth inhibition assay was performed using *Nos2*−/− (**a**) and *Gch1*−/− (**b**) BMDM cells. Macrophages were infected with 50:1 MOI BCG in the presence of 10 ng/ml IFNγ and the live BCG remaining after 96 h was enumerated by the BACTEC Mycobacterial Growth Indicator Tube (MGIT) system (MGIA), $n = 6$ per genotype *Nos2*+/+ and *Nos2*−/−, $n = 5$ *Gch1*fl/fl and $n = 5$ *Gch1*fl/flTie2cre $n = 7$ $p < 0.05$ (t-test), individual points with SEM shown. $\Delta\log_{10}CFU = \log_{10}$ CFU sample − $\log_{10}$ CFU positive control. Error bars are ±SEM. **c**, **d** Pearson correlation between *GCH1* expression and in vitro BCG growth in PBMCs of 78 BCG vaccinated South African infants. Points represent the mean of duplicate cultures. $\Delta\log_{10}CFU$ (M.BCG) = $\log_{10}$ CFU of sample − $\log_{10}$ CFU of control

performed using bone marrow isolated from adult (>10-week old) female littermate animals.

**Isolation of murine bone marrow-derived macrophages**. Bone marrow was obtained by flushing the femur and tibia of adult female mice with PBS. A single cell suspension was prepared by passing the bone marrow through a 70 µm cell strainer. Cells were then cultured in 10 cm non-tissue culture treated dishes for 7 days in DMEM:F12 (Invitrogen) supplemented with 100 U/ml penicillin and 100 ng/ml streptomycin (Sigma), 10% (v/v) fetal bovine serum (PAA Laboratories), 5 mM L-glutamine (Sigma) and 10–15% (v/v) L929 (ATCC NCTC clone 929) conditioned medium at 37 °C and 5% CO₂. The protocol for macrophage culture was validated by flow cytometry analysis of the differentiated cells using f4/80 and CD11b as macrophage cell markers.

**Blood, bone marrow and spleen leukocytes**. Single cell suspensions of splenocytes and bone marrow cells were obtained using standard protocols[58]. Blood samples were taken directly into an EDTA coated tube (Teklab). All cell populations were stained with monoclonal antibodies directed against CD11b PerCP-Cy5.5 (550993), CD4 AF647 (557681), CD8 FITC(553030), CD3 Pacific Blue (558214), B220 PE(553089), Ly-6G FITC(561105) (all BD Biosciences, 1/200) and the 7/4 antigen PE (MCA771PE, AbD Serotech, 1/50). The total cell population was gated by forward scatter and side scatter then interrogated for expression of the relevant cells markers compared to isotype controls (T-cells—CD3+, B-Cells—B220+, Neutrophils—7/4HI, Ly-6G+ or Monocytes—7/4HI, Ly-6G−). Cells were enumerated by an absolute count as described below.

**Zymosan-induced peritonitis**. Mice were injected with 1 mg zymosan (Sigma) in 500 µl PBS using an insulin syringe. After 6 h mice were lavaged post-mortem with ice-cold PBS/EDTA (5 mM), lavage fluid was injected using a 25 G needle, the

cavity mixed by agitation, and then lavage fluid was withdrawn using a 21 G needle. Cell recruitment was assessed by flow cytometry using 7/4-PE and Ly-6G-FITC using an absolute cell count method as below. Total cell recruitment was calculated as the total cells that would be recovered in the 5 ml lavage volume.

**Bacterial culture**. BCG Pasteur (ATCC 35734) was grown in-house in Middlebrook 7H9 Broth (BD, UK) supplemented with Middlebrook ADC +0.05% Tween 80. Erdman K01 (TMC107) was provided by BEI Resources (NR-15404, BEI Resources; Manassas, USA). BCG-GFP was grown under the same conditions but in the presence of Hygromycin B (50 µg/ml) to select GFP expressing bacteria.

**Flow cytometry**. All flow cytometry was performed using a DAKO CyAn cytometer and Summit software (both Beckton Coulter). Data was analysed using Flow Jo software (TreeStar Inc). Where cells were enumerated using an absolute count protocol with cells quantified by ratio to a known number of fluorescent beads spiked into the sample prior to analysis.

**Stimulation of bone marrow-derived macrophages**. Following differentiation cells were harvested and plated into 6 or 12 well plates in serum-free media (Optimem supplemented and 0.2% (w/v) low-endotoxin bovine serum albumin (Sigma)). Cells were stimulated with BCG at a MOI of 1:1 in the presence of 10 ng/ml murine IFNγ (Peprotech EC) for 24 h, parallel wells were left unstimulated. After 24 h cell pellets and cell culture supernatants were collected, or the cells subjected to biochemical analysis.

**Genomic DNA production and excision PCR**. Genomic DNA for detection of the excised allele was produced using the Qiamp kit (Qiagen). The floxed and excised allele were detected using the following primers:
    5′GTCCTTGGTCTCAGTAAACTTGCCAGG3′
    5′GCCCAGCCAAGGATAGATGCAG3′
    5′GCTCATCCCCCACACTTGTCTT3′
The floxed allele yields a 1030 bp product and the excised allele a product of 1300 bp.

**Determination of tetrahydrobiopterin levels**. BH4 and oxidized biopterins (BH2 and biopterin) were determined by high-performance liquid chromatography (HPLC) followed by electrochemical and fluorescent detection, respectively[59]. Cell pellets were freeze-thawed in ice-cold resuspension buffer (50 mM phosphate-buffered saline, 1 mM dithioerythritol, 1 mM EDTA, pH 7.4). After centrifugation at 13,200 rpm for 10 min at 4 °C, supernatant was removed and ice-cold acid precipitation buffer (1 M phosphoric acid, 2 M trichloroacetic acid, 1 mM dithioerythritol) was added. Following centrifugation at 13,200 rpm for 10 min at 4 °C, the supernatant was removed and injected onto the HPLC system. Quantification of BH4 and oxidized biopterins was obtained by comparison with external standards and normalized to protein concentration, determined by the BCA protein assay. Total biopterins (BH4 + BH2 + biopterin) were quantified where significant BH4 oxidation might be expected to occur in extracellular fluids/peritoneal lavage or unperfused whole tissue homogenates subjected to harsh homogenization for CFU quantification.

**Nitrite and NOx determination**. Nitrite accumulation was measured in samples of cell culture medium using the Griess Assay with colorimetric detection in 96-well plates. Cell culture supernatants were mixed 1:1 with the Griess reagent (Sigma) and quantified by comparison to a sodium nitrite (Sigma) standard curve produced in tissue culture media. Measurement of NOx in lavage fluid was made using a Nitrite/Nitrate kit (Cayman Chemicals, US). Analysis of NOx in lung homogenates was performed using a CLD88 NO analyzer (Ecophysics).

**Electron paramagnetic resonance**. Cultured macrophages were incubated with colloid iron (II) diethyldithiocarbamate [Fe(DETC)₂] (285 µmol/L) at 37 °C for 90 min. After incubation, cells were harvested and snap-frozen in a column of Krebs-HEPES buffer in liquid nitrogen, and EPR spectra were obtained using an X-band EPR spectrometer (Miniscope MS 200; Magnettech). Signals were quantified by measuring the total amplitude, after correction of baseline, and after subtracting background signals from incubation with colloid Fe(DETC)₂ alone.

**Dihydroethidium HPLC**. Cultured macrophages were incubated with 25 µM DHE (Invitrogen) for 30 min before being harvested for separation of DHE using gradient HPLC system (Jasco, UK) with an ODS3 reverse phase column (250 mm, 4.5 mm, Hichrom, UK) and quantified using a fluorescence detector set at 510 nm (excitation) and 595 nm (emission). Samples for injection were prepared by lysing the cell pellets in ice-cold methanol and removing protein by acid precipitation with 0.1 M HCl.

**BCG intranasal infection**. For intranasal (i.n.) infection, animals were sedated using IsoFlo (Oxford University Veterinary Services; UK) and inoculated with

$5 \times 10^6$ CFU BCG through the nostrils (25 μl/nostril). Four weeks after infection, mice were sacrificed and lung and spleen were aseptically removed.

**M.tb aerosol challenge**. Mice were challenged using a Biaera AeroMP-controlled nebuliser (Biera Technologies; Hagerstown, USA) contained in a Category Level 3 TCOL isolator. Animals were loaded in nose-only restrainers and exposed to aerosolised Erdman K01 (TMC107) (BEI Resources; Manassas, USA), prepared at a $5 \times 10^6$ cfu/ml in the nebuliser. The program was run for 10 min plus a 5 min purge with the airflow set to 12 L/min at a pressure of 20 psig. The target dose of 100–200 CFU/animal was confirmed by sacrificing two infected animals from each run 24 h post-challenge. For *M.tb.* challenge studies 8 mice per group were used. Experiments were designed to have 75% power to detect differences between groups, based on a *p* value of 0.05.

**Mycobacterial growth inhibition assay**. BMDM from all four genotypes were prepared as above. Cells were treated with IFNγ (10 ng/ml) and BCG-Pasteur with a MOI of 50:1. The tubes were incubated at 37 °C on a 360° rotator for 96 h. In parallel, a positive control was set up, by adding the same inoculum of BCG, but no cells, in a BACTEC MGIT tube (BD) supplemented with OADC/PANTA supplement (BD). After incubation, sample tubes were spun at 12,000 rpm for 10 min, and cells were lysed with 500 μl of water (Sigma) for 10 min, vortexing twice in between. The resulting solution was transferred into a supplemented MGIT and incubated until positive in a BACTEC MGIT 960 machine. Data are expressed as $\Delta \log_{10}$ CFU which is $\log_{10}$ CFU of a sample subtracted by $\log_{10}$ CFU of positive control.

**Quantification of CFU**. Lungs and spleens of infected animals were harvested after challenge. Organs were homogenised in re-inforced tubes with ceramic beads containing 1 ml PBS using Precellys 24 (Stretton Scientific, UK). Homogenised organs were diluted in PBS and dilutions were plated in Middlebrook 7H10 plates (BD), containing OADC (BD Diagnostic Systems). Plates were incubated at 37 °C and counted 3 weeks later.

**Histology and immunohistochemistry**. For TB infected lungs, histology mice were sacrificed by cervical dislocation, a slit was created in the trachea and one lung lobe placed in PBS for confirmation of CFU. A tube was inserted in the trachea and lungs were inflated with 1 ml of 10% Neutral Buffered Formalin (NBF; Sigma Aldrich, UK). The lungs were removed and placed in 10 ml of NBF overnight at 4 °C and then transferred into 70% ethanol and stored at 4 °C.

Paraffin-embedded tissue samples were sectioned (7 μm) and stained with hematoxylin and eosin (Sigma) and Acid Fast stain (BD Biosciences). Immunohistochemical staining was performed using anti-iNOS antibody (sc-651, Santa Cruz, 1/500) and anti-Galectin 3 (AF1197, R&D Systems, 1/500). This was followed by a secondary biotinylated antibody (Vector Laboratories, UK) then avidin–biotin–AP complex and visualized with Vector Red alkaline phosphatase substrate (Vector Laboratories, UK).

**Western blot analysis**. Western blot analysis was performed using antibodies against murine GTPCH (a gift from S. Gross, Cornell University New York, 1/500), iNOS (610329 BD Pharmigen, 1/1000) and β-Tubulin (ab6046, Abcam, 1/50,000), using standard protocols.

**Quantitative real-time RT-PCR**. RNA was prepared using the RNeasy kit (Qiagen) and was reverse transcribed using Superscript II (Life Technologies). 50 ng RNA equivalent cDNA was used to perform real-time PCR using pre-designed Taq-Man gene expression assays (Gch1: Mm01322973_m1, Nos2: Mm00440502_m1, TNFα: Mm00443258_m1, Life Technologies) using a BioRad CFX1000. Gene expression data was normalized to β-actin expression (4352341E, Life Technologies) using $2^{-(\text{B-actinCT-Gene of interestCT})}$.

**IL-6 ELISA**. Cell supernatants were harvested and cell debris removed by pelleting at 13,000 rpm in a microfuge. Appropriate dilutions of the supernatant media were performed and samples were analysed using an ELISAMAX Mouse IL-6 ELISA according to the manufacturer's instructions (Biolegend, UK).

**Oxidative burst**. Cells were infected with BCG-GFP/IFNγ for 2 h in antibiotic-free media supplemented with 2% FCS (to increase opsonisation) and then stained with 2.5 μM CellROX spiked into the media. After 30 min cells were rapidly detached in PBS/EDTA by scraping and cell fluorescence was analysed by flow cytometry.

**Hydrogen peroxide measurements**. Cell free supernatants were subjected to amperometric measurement of hydrogen peroxide using a Hydrogen Peroxide Microsensor electrode and Free Radical Analyser system (WPI, UK).

**Apoptosis**. Cells were infected with BCG-GFP/IFNγ for 2 h in antibiotic-free media supplemented with 2% FCS (to increase opsonisation) and then harvested

using ice-cold PBS/EDTA 5 mM. Cells were washed and stained with an 7AAD and Annexin V-APC apoptosis kit according to the manufacturer's instructions (Biolegend, UK).

**Lipid droplet formation**. Cells were infected with BCG-GFP/IFNγ overnight in antibiotic free media supplemented with 2% FCS (to increase opsonisation and as a lipid source). Cells were washed and fixed with 4% PFA for 15 min. Cells were then washed and in PBS containing LipidTOX Red Neutral Lipid Stain (Invitrogen, UK) at 1:500 dilution for a minimum of 30 min. Nuclei were costained using 1 μg/ml DAPI (Merck, UK) and cells were imaged in glass cover-slip bottomed dishes without mounting using an inverted confocal microscope. (LSM510, Carl Zeiss, Germany).

**Human PBMC MGIA**. For the mycobacterial growth inhibition assay, $1 \times 10^6$ PBMC were inoculated with ~600 CFU of BCG Pasteur in a volume of 600 μl cell culture medium (RPMI-1640 containing 10% pooled human serum, 2 mM l-glutamine and 25 mM HEPES). Samples were cultured in duplicate in 2 ml screw-cap microtubes and incubated at 37 °C for 96 h with 360° rotation. Cells were then lysed using sterile water, and the lysate transferred to a BACTEC MGIT tube with PANTA/enrichment supplement (Becton Dickinson, UK). The tubes were placed in a BACTEC MGIT 960 and incubated until positivity. The time to positivity (TTP) was then converted to $\log_{10}$ CFU using stock standard curves of TTP against inoculum volume and CFU. For each batch of samples, a direct-to-MGIT control was run in duplicate to correct for day-to-day variability in inoculum using the following calculation: $\log_{10}$ CFU of sample − $\log_{10}$ CFU of growth control.

**Human gene expression microarrays**. Microarrays on the infant PBMC samples were performed as described previously[60]. Briefly, RNA was extracted from thawed PBMCs using the RNeasy kit (Qiagen) according to manufacturer's instructions. mRNA was amplified using the Illumina Total prep kit (Ambion) according to manufacturer's instructions. RNA was labelled and hybridised to Illumina Human HT-12 v4 beadchips which were scanned on an Illumina iScan machine and data were extracted using the GenomeStudio software. Raw, probe level summary values were imported into R using beadarray[61]. Probes were background corrected using negative control probes followed by quantile normalization using the neqc command[62].

**Mouse gene expression microarray analysis**. Total RNA was extracted from each sample using a mirVANA kit (Ambion). RNA was quantified using a nanodrop spectrophotomoter and adjusted to 100 ng/ul. RNA quality control was performed using an Aglient Tape Station to confirm RNA integrity and concentration. RNA was reverse transcribed, converted into labelled cRNA, fragmented and used for hybridization. A genome wide analysis of gene expression was performed by hybridising the total RNA samples using the GeneChip WT PLUS assay for GeneTitan on the Mouse Gene 1.1 ST 16-Array plate system. The Affymetrix Genechip Command Console software 4.0 for GeneTitan was used to carry out QC checks of the procedure and produce reports.

Data underwent variance stabilization and normalization (VSN) algorithm using Limma[63]. Normalized data were imported to Bioconductor (R version 2.14) (Biobase) and comparison by genotype was performed for each condition (infected and control) to identify significantly differentially expressed genes passing a false discovery rate of $< 0.05$[64]. The effect of infection was confirmed by comparison within the wildtype animals for infected vs control. The lists of differentially expressed genes were then subject to IPA pathway analysis (Ingenuity Systems, http://www.ingenuity.com). Due to the large number of significantly regulated genes for the effect of infection the top 2538 most significant genes were used for IPA analysis ($p < 1 \times 10^{-7}$). For analysis of the effect of loss of *gch1* or *nos2* those genes passing adj $p < 0.05$ were used. Analysis of gene function or upstream regulators was performed and categories with *Z* score $> 2/-2$ (allowing directionality to be determined) and an overlap *p* value $< 0.05$ were considered significant.

For Gene Set Enrichment Analysis (GSEA) preranked lists for the effect of genotype within infected samples were created. GenePattern v3.9.10 (Broad Institute) was used and lists were subjected to analysis the GSEAPreranked module using the GO biological processes gene sets (C5 BP) and Curated Pathways (C2 CP) from the Molecular Signature Database v6.0 (MSigDB)[65–67]. Gene Sets with a FDR $q < 0.1$ and *p* value $p < 0.005$ were considered significant in this analysis and used in further analyses. Network analysis of significantly regulated gene sets ($p < 0.005$ and FDR $q < 0.1$) was performed using Cytoscape v3.5.1 and the Enrichment map plugin[30,68,69], the Enrichment Map default overlap coefficient of 0.5 was used, as described in ref. [30]. To create a list of gene sets unique to the loss of *Gch1* any genes passing $p < 0.005$ and FDR $q < 0.1$ present in both *Gch1* and *Nos2* deficient macrophage analyses were removed from the *Gch1* list and the remaining gene sets replotted in Cytoscape.

**Quantification and statistical analysis**. Statistical significances are indicated in figures (*$p < 0.05$) and mentioned in the figure legends. Data were statistically significant when $p < 0.05$. Statistical tests used are indicated in the figure legends

and were calculated with Graph Pad Prism 6. Venn Diagrams were produced using BioVenn.nl[70]. Indicated $n$-values in figure legends represent individual biological replicates, i.e. individual mice or cell culture derived from different individuals. Bar chart and individual point data are expressed as mean ± SEM, unless stated in the figure legends. Comparisons between WT and $GCH1^{fl/fl}$Tie2cre were made by two-way ANOVA with Bonferroni post-tests between individual groups or two-tailed $t$-test (2 sample groups). Where two sample groups were compared by unpaired $t$-test an $F$ test was also performed to compare variance, where this indicated that variances were significantly different Welch's corrected $t$-test was used. In this study Welch's correction was applied to gene expression data. Time course studies were compared by ANOVA for repeated measurements. CFU studies were analysed by Mann–Whitney test, with statistical outliers identified using Grubb's test and removed from the dataset. In this case a single value was removed from the spleen CFU values in Fig. 2. Correlation data was analysed using Pearson's correlation coefficient within Prism 6, both $r^2$ and $p$ value are reported.

## Data availability

For the human studies, raw and normalized expression data have been deposited at Gene Expression Omnibus under the accession number GSE98550. For the mouse studies, raw and normalized expression data have been deposited at GEO under the accession number GSE107543. Source data for western blot and DNA gels and flow cytometry gating in Figures 1 and 2 is available in the supplementary methods file. Full lists of genes and genesets passing significance thresholds are provided as supplementary datasets. A reporting summary for this article is available as a Supplementary Information file.

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

## Acknowledgements

This work was supported by the British Heart Foundation [RG/15/10/31485], [RG/17/10/32859] and [CH/16/1/32013], Wellcome Trust [095780/Z/11/Z] and [203141/Z/16/Z], The John Fell Fund, The Oxford University Medical Research Fund, and the National Institute for Health Research (NIHR) Oxford Biomedical Research Centre. BCG-GFP was a gift from Rajko Reljic from St. George's University of London (SGUL).

## Author contributions

Conceptualization: E.M.N., E.S., H.M.S., and K.M.C. Methodology: E.M.N. and E.S. Formal analysis: H.L. and J.M. Investigation: E.M.N., E.S., M.J.C., R.H.-K., R.T., M.K.S., M.M., A.-L.K., A.B.H., and M.D. Resources: P.B., H.A.F., and D.R.G. Vizualization: E.M.N. and H.L. Writing—original draft: E.M.N., E.S., H.M.S., and K.M.C. Writing—review and editing: E.M.N., E.S., D.R.G, H.M.S., and K.M.C. Funding acquisition: E.M.N., D.R.G., H.M.S., and K.M.C.

## Additional information

**Competing interests:** The authors declare no competing interests.

