## [Peer Review File · Nature Communications]

Reviewers' comments:

Reviewer #1 (Remarks to the Author):

This paper from Dr McNeil and her colleagues is an interesting study that cleverly dissociates the specific roles of inducible nitric oxide and tetrahydrobiopterin in resistance to the TB bacterium. This was achieved using a conditional Gch1 knockout mouse model. The studies were nicely done, and convincing.

A few comments:

Fig.3a. The histology slides are very small and hard to evaluate [especially the lung sections].

P11. Fig-7, I assume they mean Fig-5. The final paragraph here seems to be rather stating the obvious, not helped by the overall section. I don't know of many people who will even understand any of this – it needs to be spelled out more simply for us not deeply versed in gene mapping – not helped by the fact that the figure looks like a set of star charts.

The Discussion is very lengthy and perhaps could be shortened somewhat.

My main criticism regards the introduction and subsequent interpretation of the data. The role of NO in the infected lungs began to be determined by two studies [in 1997 and 2000] in which NOS2-KO mice were shown to be susceptible to high dose intravenous infections [with TB and BCG]. A 2001 paper, cited here, found that this also applied to low dose aerosol. What is not cited here however is the first demonstration of this using this infection route by Cooper [IAI 2000]; moreover, the title of her paper directly includes the statement that NO is "not essential", exactly the point the current paper is making. Those who do not study history are doomed to repeat it.

There are clearly other mechanisms involved, as the authors argue. They might also look at Correa et al [IAI 2014], who looked at the endothelin system, the homeostatic counterpart of the NO system.

Reviewer #2 (Remarks to the Author):

The paper by McNeill et al entitled 'Regulation of Mycobacterial Infection by Macrophage Gch1 and Tetrahydrobiopterin' reports on the consequences of deleting the GTP cyclohydrolase-1 (GTPCH-1)-encoding gene (gch-1) in leukocytes in mice. GTPCH-1 is one of the enzymes required for the generation of the NOS cofactor tetrahydrobiopterin (BH4), that is an essential cofactor for nitric oxide (NO) production. Thus, deleting gch-1 allows to dissect the role of iNOS and NO in the control of mycobacterial infections.

The authors describe that such deletion results in a significant depletion of BH4, both in unstimulated as well as BCG/IFN γ -stimulated macrophages. Furthermore, they show that both BCG as well as *M. tuberculosis* growth is equally or even better controlled in the mutant mice as compared to wild type mice. This is in contrast to iNOS-deficient mice that rapidly succumb to an infection with *M. tuberculosis*. Finally, the authors perform gene expression analysis and document a number of deregulated genes upon gch-1 deletion, both under uninfected as well as BCG/IFN-g-stimulated conditions.

Tuberculosis continues to be an important problem world-wide, and knowledge of the mechanisms that are involved in the control of its causative agent, *M. tuberculosis*, are important to decipher. In that sense, the work by McNeill et al showing enhanced protection of mice lacking gch-1 is

interesting, especially in the light of abundant evidence of the importance of iNOS, presumably via the induction of NO to mycobacterial control.

In that respect, the data showing that mice lacking *gch-1* can control an *M. tuberculosis* infection similar to wild type mice, while *Nos2*^{-/-} mice rapidly succumb to the infection are highly interesting. There are however a number of issues with the manuscript. First, the mechanisms via which control is achieved remains enigmatic; it is well known that depletion of BH4 increases reactive oxygen production, and while this would be an interesting mechanisms to further analyze the authors only mention it as a possibility to explain their data. Second, all the mycobacterial infection studies including the gene expression analysis are performed by including IFN γ , without any justification to do so, and which appears to bias many of the results shown. Third, several of the data are repetitions of work already published (McNeill et al., 2015, *Free Radical Biology and Medicine*79:206–216).

Specific points

1. A major concern is the inclusion of IFN-g throughout most of the experiments, even though this is not stated at all in the Result section (that mentions merely 'BCG-infected' throughout the manuscript, see page 5 and following). Stimulation of macrophages with interferon-g is such a strong activation stimulus, that it is likely to modulate multiple genes independently as compared to an infection with BCG and/or *M. tuberculosis* alone. There is no evidence presented in the paper whether infection with mycobacteria alone would have similar effects as compared to the BCG/IFN-g combination. Furthermore, while it is well known that mycobacteria suppress interferon-g responses (see e.g. Ting et al. *Jl* 1999, 163: 3898; Nagabhushanam et al., *Jl* 2003, 171:4750 and Fortune et al., *Jl* 2004, 182:6272), even in the experiment displayed in Fig. 2i the authors add IFN-g.
2. The data in Figs 2c,d and Fig 3 suggest that *Nos2* confers resistance towards mycobacterial infections via NO-independent mechanisms. While these are interesting observations, one possible explanation would be the observed upregulation of reactive oxygen species upon *gch-1* deletion (as reported in McNeill et al., 2015, *Free Radical Biology and Medicine*79:206–216 as well as in this manuscript) but this is not further explored.
3. Instead, the authors embark on a gene expression analysis and highlight a number of altered pathways, but there are several issues with this analysis. First, given the presence of IFN-g in all samples containing BCG it is impossible to dissect the consequences of mycobacteria from IFN-g activation. Second, for none of the pathways mentioned (lysosomal function, cell survival, cellular metabolism) there are any experiments shown to analyze their possible involvement in mycobacterial control in a *gch-1*-dependent manner.
4. The authors fail to find a correlation between iNOS expression and BCG growth in human samples (Fig. 6), even though they cite on p. 3 the work from Gomez et al. on the association of iNOS polymorphism with tuberculosis but no further discussion is provided.
5. Several of the data have been published by the same group in McNeill et al., 2015, *Free Radical Biology and Medicine*79:206–216; this holds through for Fig1a, c-e and Fig 2a-b and therefore are redundant here. Also the gene expression analysis of *gch-1* versus wildtype mice have been published in the aforementioned paper, revealing 78 genes whose expression was altered between *gch-1* and wild type macrophages. No mentioning of the 2015 dataset is made here; are the 77 genes mentioned in the present manuscript (second paragraph p. 10) overlapping? Is this the same dataset re-analyzed?

Other points

1. In Fig 1k and Fig 2f,g the authors highlight the differences between total biopterins in the wild type and mutant cells, but isn't the difference in BH4 levels important? Also, is the difference between wild type and mutants not mainly reflecting the difference in BH4 levels?
2. While for BCG infections lung CFU's are similar for wild type and *gch-1*^{-/-} mice and spleen CFU's

reduced in the *gch-1*^{-/-} mice, this is reversed for the *M. tuberculosis* infection shown in Fig. 3, without providing any explanation.

3. All blots should be labeled with the appropriate molecular weight markers.

4. The mouse data shown reflect different group sizes, especially in Figs 2c,d (wild type mice $n=10$; mutant mice $n=7$ (2c) and $n=6$ (2d)) without any explanation as to why these mice numbers are different.

5. Page 11 line 7: Figure 7 should be Fig 5.

Reviewers' comments:

Reviewer #1 (Remarks to the Author):

This paper from Dr McNeil and her colleagues is an interesting study that cleverly dissociates the specific roles of inducible nitric oxide and tetrahydrobiopterin in resistance to the TB bacterium. This was achieved using a conditional *Gch1* knockout mouse model. The studies were nicely done, and convincing.

We thank the Reviewer for their very constructive assessment of the manuscript. We have addressed all of the Reviewer's comments, with responses below.

A few comments:

Fig.3a. The histology slides are very small and hard to evaluate [especially the lung sections].

We accept the Reviewer's comment that the histology images become too small once the figure is resized to a manuscript format, with the accompanying loss of resolution. We have now reformatted the figure to include higher magnification views of the gross morphology. To further expand the histology information, beyond the panels presented in the Figure 3, we now include an additional Supplementary Figure incorporating large-format tiled images of the whole lung and additional immunohistochemistry to demonstrate macrophage (Galectin 3/mac2 epitope) infiltration into the lung granuloma in the Fig 3a. These new panels, and new Figure, add substantially to the demonstration of pathology in the *Gch1* and *Nos2* knockout mouse lungs, in response to mycobacterial infection.

il

Figure 3: Absence of *Gch1* and BH4 increases resistance of mice to virulent *M.tb* infection. *Gch1^{fl/fl}Tie2cre* (red) and wildtype (black), *Nos2^{-/-}* (green) and *Nos2^{+/+}* (blue) mice were infected with 150 CFU *M.tb* (Erdman strain) by aerosol inhalation. 23 days following infection the lungs and spleen were removed for analysis. Representative organs were imaged from each genotype prior to n=2 tissues per genotype being embedded for histochemical analysis. Sections were cut from all tissues and stained with H&E or subjected to anti-iNOS immunohistochemistry (red) (a).

Gch1^{fl/fl}

Gch1^{fl/fl}Tie2cre

Nos2^{+/+}

Nos2^{-/-}

Supplementary Figure 5: Lung gross morphology. Mice were infected with 150 CFU *M.tb* (Erdman strain) by aerosol inhalation. 23 days following infection the lungs were removed for analysis. Representative organs were imaged from each genotype prior to n=2 tissues per genotype being embedded for histochemical analysis. Sections were cut from all tissues and stained with H&E. Tiled images of the whole lung were produced.

Supplementary Figure 6: Macrophage localisation to granuloma. Mice were infected with 150 CFU *M.tb* (Erdman strain) by aerosol inhalation. 23 days following infection the lungs and spleen were removed for analysis. Representative organs were imaged from each genotype prior to n=2 tissues per genotype being embedded for histochemical analysis. Sections were cut from all tissues and subjected to anti-Galectin 3 (mac2 epitope) immunohistochemistry to detect macrophages within granuloma (Galectin 3 – red, nuclei - blue).

We have referred to this new data in the manuscript, as follows:

“Immunohistochemistry of lung granuloma confirmed iNOS expression and macrophage infiltration within the granulomata of wildtype and *Gch1^{fl/fl}Tie2cre* mice that was absent in *Nos2^{-/-}* mice (Figure 3a and Supplementary Figure 6). ”

P11. Fig-7, I assume they mean Fig-5. The final paragraph here seems to be rather stating the obvious, not helped by the overall section. I don't know of many people who will even understand any of this – it needs to be spelled out more simply for us not deeply versed in gene mapping – not helped by the fact that the figure looks like a set of star charts.

We are grateful to the Reviewer for picking up the incorrect reference to Figure 7, which we have now corrected to Figure 5. We accept the Reviewer's comment that the GSEA section of the manuscript could be improved to make the gene set data more concise and direct in style, and that by including all the accompanying tables as supplementary material does not make the information accessible for the reader. In accordance with the Reviewer's comment, we have now added a brief table to accompany this Figure, summarising the main pathways highlighted in the text. We have also added a legend explaining the gene set enrichment map figure.

We accept that we had not clearly described the format of the presentation of the gene set data in the original manuscript. The presentation of the gene set pathways using the format very appropriately described by the Reviewer as a 'star chart' is often termed a 'constellation map'. This format is a standard approach for analysing and presenting the gene sets and pathways identified from transcriptomic experiments. We followed the approach exemplified by WN Haining et al. from the Broad Institute, who published several high profile papers using 'Constellation Map' (e.g. Akondy *et al*, Nature 2017;), although we used Cytoscape as a more updated package to generate and annotate the gene set data.

For the Reviewer's information, both the GSEA analysis tools and the Cytoscape software and plugins are freely available and supported by explanatory information, which we have now referenced more clearly ([http:// genepattern.broadinstitute.org](http://genepattern.broadinstitute.org) and <http://www.cytoscape.org/>).

In addition, we have now performed additional analyses to validate the gene expression results, demonstrating alterations in macrophage function that map to the exemplar genesets (please see response to Reviewer 2, below). These new data have enabled us to make more direct comment on the interpretation of the gene expression datasets.

We have altered the text and figure in this section, as follows:

“However, number of gene signatures were unique to the loss of BH4 (Figure 5C and D). These clusters related fundamental cellular processes such as metabolism and cellular detoxification as well as to processes related to inflammation and infection such as lysosomal function, apoptosis and cell senescence and innate immune. Taken together, the IPA and GSEA data indicate that macrophage *Gch1* and tetrahydrobiopterin are specifically associated with regulation of the innate inflammatory response, cellular metabolism and mitochondrial function, cell detoxification and cell survival. To confirm if the gene expression changes correlated with alterations in cell function we tested whether macrophage inflammatory activation and cell survival were altered, as is indicated by the gene expression analysis. *Gch1*^{fl/fl}Tie2cre macrophages showed reduced IL-6 gene and protein expression in response to BCG/IFN γ , as predicted (Supplementary Figure 11a). Under the conditions used for gene expression analysis no alterations in cell viability were seen, under conditions with similar levels of infection of the bulk population (Supplementary Figure 11b). Under conditions of enhanced greater stress, with cells being infected with BCG/IFN γ (MOI 5:1) we observed a decreased macrophage cell survival in the absence of *Gch1* again confirming that the gene expression analysis is predictive of alterations in macrophage cell biology.”

Figure 5: Gene set enrichment using GO term functional annotations identifies shared and unique gene expression patterns in BCG/IFN γ treated macrophages. Enriched gene sets in BCG infected macrophages, identified by GSEA (MSigDb, C5 Go annotations set). Gene sets passing FDR $p < 0.1$ with an overlap coefficient 0.5 were used to plot an Enrichment Map in Cytoscape. Individual enrichment maps were plotted for infected **(a)** *gch1* and **(b)** *nos2* macrophages and clusters were annotated by hand. To determine how far regulated gene functions were shared between the *gch1* and *nos2* knockout macrophages the significantly regulated gene sets were compared and genes only in the *gch1* network **(c)** were plotted and the clusters annotated. **(d)** A table summarising the GO-term clusters, with exemplar GO terms.

Figure 5

The Discussion is very lengthy and perhaps could be shortened somewhat.

We accept the Reviewer's comment that the Discussion could be shortened. In addition to the changes already made in the Results sections, described above, we have also edited the Discussion to be more concise and readable. We have removed 3 large paragraphs from the discussion and have moved our headline interpretation to the start of the discussion.

My main criticism regards the introduction and subsequent interpretation of the data. The role of NO in the infected lungs began to be determined by two studies [in 1997 and 2000] in which NOS2-KO mice were shown to be susceptible to high dose intravenous infections

[with TB and BCG]. A 2001 paper, cited here, found that this also applied to low dose aerosol. What is not cited here however is the first demonstration of this using this infection route by Cooper [IAI 2000]; moreover, the title of her paper directly includes the statement that NO is “not essential”, exactly the point the current paper is making. Those who do not study history are doomed to repeat it.

We are grateful to the Reviewer for pointing out these important previous studies, which we had not included in our citations of the NOS2-KO literature in the original manuscript. We have now carefully reviewed these studies and have incorporated their findings in to the Introduction and the Discussion.

We agree with the Reviewer that there is some variability in the timing of the loss of control of infection in the models of mycobacterial infection used to test the susceptibility of the *Nos2*^{-/-} mice. Scanga *et al.* showed that infection with either 10⁵-10⁶ cfu intravenously, or 50 cfu by aerosol, is controlled for up to 15 days in *Nos2*^{-/-} mice, but progresses thereafter, compared with WT mice. Cooper *et al.* found that low dose aerosol infection (100 viable bacteria) was controlled in *Nos2*^{-/-} mice for up to 30 days, but by 50 days was not controlled in the lung (Figure 1 of Cooper *et al.*), and with gross evidence of progressive lung, liver and spleen enlargement by 50 days (Table 1). In these and all other studies, the *Nos2*^{-/-} mouse demonstrates increased susceptibility to mycobacterial infection as time progresses. As the precise dose and route of exposure influences the timepoint at which susceptibility is seen in the *Nos2*^{-/-} mouse, we specifically included the *Nos2*^{-/-} mouse in our studies so that we could confidently test the phenotype of the *Gch1*^{fl/fl}Tie2cre mouse in comparison with both the wild type and *Nos2*^{-/-} mice. In Cooper *et al* mice were infected with approximately 10⁵ CFU at 30 days. In comparison, using our challenge method with a dose of 150 CFU, *Nos2*^{-/-} mice had to be culled at 21 days post infection, as they had reached a humane end point, with 10⁷ CFU lung bacterial load. Our challenge system therefore appears to be more sensitive at detecting differences between *Nos2*^{-/-} and *Nos2*^{+/+} at earlier time points. Importantly, *Gch1*^{fl/fl}Tie2cre mice appeared to be more resistant using this stringent infection model.

In accordance with the Reviewer’s important comments, we have now included the additional citations in the manuscript, and discussed these points in the introduction and results, as follows:

“Progress in the identification of genetic factors involved in TB susceptibility has been limited but the importance of some genes is already clear². Polymorphisms in inducible nitric oxide synthase (iNOS, encoded by *NOS2*)³ confer important differences in the control of mycobacterial infection. Mice deficient in iNOS show variability in the timing of loss of control of mycobacterial infection in experimental models, however, in all published studies to date *Nos2*^{-/-} mice show increased susceptibility at later time points. *Nos2*^{-/-} mice succumb to infection with intravenous *M. Bovis* Bacillus Calmette Guerin (BCG) after 8-12 weeks, in contrast to wildtype mice that are able to control infection⁴. Furthermore, iNOS-deficient mice infected with the more virulent *M.tb* Erdman succumb more rapidly than control mice in both intravenous and aerosol challenge experiments, even if early control of infection is maintained in short challenges^{5,6,7}. Thus, iNOS is a pivotal regulator of susceptibility to *M.tb* infection in mice. These effects of iNOS have been associated with the production of nitric oxide (NO), largely because of *in vitro* correlations between the mycobacteriocidal activity of activated murine macrophages and the amount of reactive nitrogen intermediates (RNI) produced^{8,9} and the direct cytotoxicity of nitric oxide to mycobacteria¹⁰.”

And

“Inclusion of this control group was of particular importance given the variability in onset of susceptibility seen in previous studies^{5,6,7}. ”

There are clearly other mechanisms involved, as the authors argue. They might also look at Correa et al [IAI 2014], who looked at the endothelin system, the homeostatic counterpart of the NO system.

We thank the Reviewer for highlighting this study, and the suggestion to consider the ET-1 system in TB infection. The Reviewer's comment is particularly relevant, as both the endothelin 1 (ET-1) and endothelin receptor B (ETB) genes appear in the list of genes specifically regulated by *Gch1* in infected macrophages. Correa et al. combined endothelin receptor A & B inhibitors and showed an increase in disease in wildtype and *Nos2*^{-/-} animals, suggesting that the action of the ET-1 system may act independently of iNOS expression. ET-1 treatment decreased inflammation, but with no effect on bacterial load. ETB antagonism alone caused increased bacterial load with no increase in lesion area, whereas selective ETA antagonism causing increase inflammation and no increase in bacterial load. In our study, loss of *Gch1* was associated with an increase in ET-1 expression and a decrease in ETB expression. The overall biological effect of this combination of changes in ET-1 and ETB is difficult to predict, and we would respectfully suggest that new experiments to specifically investigate the role of the ET-1 system are outside the scope of the current experiment. Nevertheless, we thank the Reviewer for this comment and we have included additional points in the Discussion addressing the expected observation that reduced NO in *Gch1*-deleted cells is associated with reduced ET-1 gene expression, but also with ETB receptor downregulation, without a clear expectation of an effect on bacterial load, based on the published literature.

“The endothelin system is a key effector of NOS enzymes in other cell types and also showed alterations in gene expression levels in the absence of BH4, but expression of both protective (ET-1) and non-protective (ETB) elements of the system were downregulated⁴³. ”

Reviewer #2 (Remarks to the Author):

The paper by McNeill et al entitled 'Regulation of Mycobacterial Infection by Macrophage Gch1 and Tetrahydrobiopterin' reports on the consequences of deleting the GTP cyclohydrolase-1 (GTPCH-1)-encoding gene (*gch-1*) in leukocytes in mice. GTPCH-1 is one of the enzymes required for the generation of the NOS cofactor tetrahydrobiopterin (BH4), that is an essential cofactor for nitric oxide (NO) production. Thus, deleting *gch-1* allows to dissect the role of iNOS and NO in the control of mycobacterial infections.

The authors describe that such deletion results in a significant depletion of BH4, both in unstimulated as well as BCG/IFN γ -stimulated macrophages. Furthermore, they show that both BCG as well as *M. tuberculosis* growth is equally or even better controlled in the mutant mice as compared to wild type mice. This is in contrast to iNOS-deficient mice that rapidly succumb to an infection with *M. tuberculosis*. Finally, the authors perform gene expression analysis and document a number of deregulated genes upon *gch-1* deletion, both under uninfected as well as BCG/IFN-g-stimulated conditions.

Tuberculosis continues to be an important problem world-wide, and knowledge of the mechanisms that are involved in the control of its causative agent, *M. tuberculosis*, are important to decipher. In that sense, the work by McNeill et al showing enhanced protection of mice lacking *gch-1* is interesting, especially in the light of abundant evidence of the importance of iNOS, presumably via the induction of NO to mycobacterial control.

In that respect, the data showing that mice lacking *gch-1* can control an *M. tuberculosis* infection similar to wild type mice, while *Nos2*^{-/-} mice rapidly succumb to the infection are highly interesting. There are however a number of issues with the manuscript. First, the mechanisms via which control is achieved remains enigmatic; it is well known that depletion of BH4 increases reactive oxygen production, and while this would be an interesting mechanisms to further analyze the authors only mention it as a possibility to explain their data. Second, all the mycobacterial infection studies including the gene expression analysis are performed by including IFN γ , without any justification to do so, and which appears to bias many of the results shown. Third, several of the data are repetitions of work already published (McNeill et al., 2015, *Free Radical Biology and Medicine* 79:206–216).

We thank the Reviewer for their detailed assessment of our manuscript and their enthusiasm for our findings. The observation that the *Gch1*-deficient mouse is not more susceptible to TB, and indeed shows protection against TB, and evidence of increased TB killing, is striking since the *Nos2*^{-/-} mouse is highly susceptible. As the Reviewer points out, this implicates signalling effects related to iNOS and BH4, but raises the possibility that there are new mechanisms relating iNOS to TB susceptibility that are not dependent on NO, since neither the *Gch1* nor *Nos2* knockouts generate iNOS-derived NO, whereas only the *Nos2* knockouts are more susceptible to TB. We have now produced additional data to support our findings and experimental design and have made substantial changes to the manuscript text in answering the Reviewer's comments. We have detailed the new experimental data and manuscript changes in the point-by-point section, below.

Specific points

1. A major concern is the inclusion of IFN-g throughout most of the experiments, even though this is not stated at all in the Result section (that mentions merely 'BCG-infected')

throughout the manuscript, see page 5 and following). Stimulation of macrophages with interferon-g is such a strong activation stimulus, that it is likely to modulate multiple genes independently as compared to an infection with BCG and/or *M. tuberculosis* alone. There is no evidence presented in the paper whether infection with mycobacteria alone would have similar effects as compared to the BCG/IFN-g combination. Furthermore, while it is well known that mycobacteria suppress interferon-g responses (see e.g. Ting et al. JI 1999, 163:3898; Nagabhushanam et al., JI 2003, 171:4750 and Fortune et al., JI 2004, 182:6272), even in the experiment displayed in Fig. 2i the authors add IFN-g.

We accept the Reviewer's comment that the use of the BCG/IFN γ combination in the *in vitro* experiments was not clearly explained or justified in the original manuscript. We established the *in vitro* model to test the mechanisms related to the *in vivo* observations, where infection with BCG or *Mtb* leads to robust iNOS expression (as exemplified in Figure 3 and Supplementary Figures 6 and 7). Accordingly, we examined iNOS expression and nitric oxide production by bone marrow derived macrophages (BMDM), but found (as others have) that infection with mycobacteria alone in cultured BMDM was not sufficient to stimulate detectable nitrite accumulation and that activation of *Nos2* gene expression was minimal (see Figure, below, now included in new Figure 1). This is in keeping with the published literature that demonstrates *Nos2* to be a key gene induced by IFN γ (Flynn JL et al JEM 200). BCG in combination with IFN γ caused a robust induction of *Nos2* and nitrite accumulation. Thus, in BMDM *in vitro*, IFN γ is required as a co-stimulus to induce efficient iNOS expression; if we did not induce iNOS expression *in vitro* then we would be unable to test the effect of BH4 and iNOS on *Mtb* infection in BMDM, as a model system relevant to the *in vivo* observations.

We emphasise that in all *in vivo* studies, iNOS expression was not induced by additional cytokine or IFN γ , but resulted solely from mycobacterial infection. As shown in the immunohistochemistry panels in Figure 3, iNOS is clearly present in the lung granuloma of infected mice (but not in *Nos2* knockouts), and it is in the *in vivo* models where the striking phenotype is observed.

We accept that we were not consistent in the labelling of the *in vitro* data between Figure 1 and Figure 4 and 5. We have now changed the figures and text to describe the experimental conditions as untreated (UT) or treated with BCG/IFN γ . To be clear about the justification for this experimental design we have now included a demonstration of the requirement for IFN γ to induce iNOS in Figure 1 (see below) with direct reference to this finding in the text.

Excerpt of Figure 1 legend: (a) Measurement of *nos2* and nitrite accumulation in bone marrow derived macrophages stimulated BCG (MOI 1:1) or IFN γ (10ng/ml) and BCG (MOI 1:1) in wildtype *Gch1f/fl* macrophages. Error bars are \pm sem, ANOVA with Bonferroni post-test *P<0.05, **P<0.01, ***P<0.001).

” *In vitro* infection of macrophages with BCG (Pasteur) alone showed minimal induction of *Nos2*, however in the presence of IFN γ infection with BCG caused a robust induction of *Nos2* and accumulation of nitrite, indicating robust iNOS NO-producing activity (Figure 1a). The specificity of nitrite accumulation for iNOS activity was confirmed by infection of *Nos2*^{-/-} BMDM under identical conditions (Figure S2). BCG/IFN γ treatment caused increased cellular BH4 content in wildtype mice, with BH4 levels significantly reduced to near undetectable levels in the *Gch1*^{fl/fl}Tie2cre cells. Loss of BH4 did cause affect induction of iNOS protein (Figure 1b), but prevented iNOS NO-producing activity measured as nitrite accumulation, arginine-citrulline conversion and direct detection of authentic NO generation by EPR (Figure 1 d-f). In order to test whether iNOS protein had other non NO-producing roles in mycobacterial function the BCG/IFN γ combination was used in all *in vitro* studies to cause robust iNOS expression. ”

2. The data in Figs 2c,d and Fig 3 suggest that *Nos2* confers resistance towards mycobacterial infections via NO-independent mechanisms. While these are interesting observations, one possible explanation would be the observed upregulation of reactive oxygen species upon *gch-1* deletion (as reported in McNeill et al., 2015, Free Radical Biology and Medicine79:206–216 as well as in this manuscript) but this is not further explored.

We are very grateful to the Reviewer for emphasising this point. We have previously reported evidence of altered ROS signalling in *Gch1*-deficient macrophages in our 2015 *FRBM* paper. However, in this previous paper mycobacterial infection was not studied, and the changes in redox biology in the BH4 deficient macrophages is complex. In accordance with the Reviewer’s comments, we have now addressed the following points:

- (1) We observed significantly elevated production of ethidium from dihydroethidium, in *Gch1*^{fl/fl}Tie2cre BMDM under all conditions (UT and BCG/IFN γ), indicating that this response is not dependent on either iNOS or mycobacterial infection. The nature of the ethidium signal generated from dihydroethidine is uncertain, with ethidium production not being linearly related to superoxide concentration (unlike the 2-hydroxyethidium signal), and shown to be related to multiple sources including hydrogen peroxide, metal or heme-containing reactions or peroxidase activity (Fernandes et al AJPCP 2007) . So although the production of ethidium demonstrates a clear ROS-dependent phenotype, the precise nature of the ROS involved remains unclear.
- (2) To investigate changes in ROS production using alternative techniques, in infected macrophages, we have now performed new experiments using a flow cytometry (FACS) assay, adapted from Eckelt *et al* (2015). We infected macrophages with GFP-expressing BCG (GFP-BCG) in the presence of IFN γ , to maintain similar conditions to our existing studies, for two hours prior to staining of the cells with the CellROX ROS indicator for 20min. The use of GFP-BCG allows FACS analysis of the specific population of infected (GFP+) macrophages, compared with non-infected macrophages (GFP-), within the same sample (Figure, below). An MOI of 5:1 was used to allow sufficient GFP+ cells to be analysed. When compared to GFP- cells within the same population, GFP+ macrophages had significantly higher ROS production by two-way ANOVA. However, no significant difference in ROS production between *Gch1*^{fl/fl} and *Gch1*^{fl/fl}Tie2cre macrophages was observed. These data indicate that the initial oxidative burst following mycobacterial infection in macrophages is preserved.

Gch1^{fl/fl}Tie2cre and *Gch1^{fl/fl}* macrophages were stimulated IFN γ (10ng/ml) and BCG (MOI 5:1) for 2 hours. CellROX cell permeable ROS indicator dye was added to the media at 500nM for an additional 30min. At the end of the assay the cells were washed twice, harvested and analysed by flow cytometry within 120mins. GFP+ cells were gated by comparison to non-infected controls and the Cell ROX signal quantified in the GFP+ and GFP- populations. Tertbutyl hydroperoxide (THBP) was included as a positive control to induce oxidative stress. Data was analysed by 2-way ANOVA * p<0.05. No significant effect of genotype was detected by post-testing (n=7/genotype).

(3) To obtain a further profile of the ROS phenotype of *Gch1fl/flTie2cre* macrophages, we next assessed hydrogen peroxide production. Using an amperometric method for quantification of H₂O₂ in cell culture medium we did not detect any extracellular H₂O₂ in media from untreated cells, but abundant H₂O₂ was detected when wildtype macrophage were infected with BCG/IFN γ (MOI1:1). This signal was significantly lower in supernatants from BH4-deficient macrophages. These data together show that multiple ROS sources, and ROS fates, are altered in BH4-deficient macrophages.

H₂O₂ accumulation in the cell culture media over 24hrs of cell stimulation with BCG/IFN γ from *Gchf1f/fl* and *Gch1fl/flTie2cre* bone marrow derived macrophages was measured using a hydrogen peroxide electrode (n=4/genotype). * P<0.05 two-tailed T-test.

Taken together, our data on ROS production and release by BH4 deficient macrophages reveal a complex picture, where individual sources and/or fates of ROS are elevated, decreased or unchanged. This finding implies that ROS production from multiple sources are likely altered in the absence of BH4. This fits with the complex role of ROS in control of mycobacterial growth. The well documented mycobacterial downregulation of ROS is an important component of the immune evasion mechanisms, necessary for the persistence of this successful pathogen. Yet inflammation, mediated in part by ROS, is critical for TB pathogenesis.

We have addressed this in the text as follows:

“Whilst nitric oxide production was absent, production of ROS was significantly increased in *Gch1^{fl/fl}*-Tie2cre macrophages both at baseline (1.6 fold) and following BCG infection (3.2 fold) when assessed by the production of ethidium from dihydroethidium (Figure 1g). This indicates that the loss of BH4 causes alterations in cellular redox state in both the presence and absence of iNOS. Macrophages have multiple potential sources of ROS, including the phagocytic NADPH oxidase that mediate oxidative burst and mitochondrial metabolism. To assess more broadly how ROS production by infected macrophages was affected the phagocytic oxidative burst was assessed by flow cytometry using the CellROX indicator dye²⁷. BMDM were infected with BCG-GFP²⁸ and the ROS production was assessed in infected BCG-GFP+ vs uninfected BCG-GFP- cells within the same population 2 hours after initial infection with BCG-GFP/IFN γ . Whilst a significant induction of ROS production was observed, this was not significantly affected by loss of BH4 (Supplementary Figure 3a). To further examine redox changes in response to infection we measured the extracellular release of H₂O₂, which is induced by infection and was undetectable in uninfected cells, this was significantly reduced in the *Gch1^{fl/fl}*-Tie2cre cells (Supplementary Figure 3b) indicating that the changes in redox biology elicited by the loss of BH4 in macrophages is substantially more complex than has been observed in other BH4 deficient cell types²⁹. Whilst in endothelial cells ROS production is elevated as a result of eNOS uncoupling, the greater variety of sources and potency of ROS production by macrophages results in more widespread changes, but with the phagocytic oxidative burst in response to infection appearing unchanged.”

3. Instead, the authors embark on a gene expression analysis and highlight a number of altered pathways, but there are several issues with this analysis. First, given the presence of IFN-g in all samples containing BCG it is impossible to dissect the consequences of mycobacteria from IFN-g activation.

Please see earlier comment.

Second, for none of the pathways mentioned (lysosomal function, cell survival, cellular metabolism) there are any experiments shown to analyze their possible involvement in mycobacterial control in a *gch-1*-dependent manner.

We accept that the gene expression studies would be strengthened by experiments to test the relevance to potential biological pathways.

We first validated the expression of genes that show differential regulation in BH4-deficient cells. As shown below, *As3mt* is significantly altered only in BH4-deficient cells (upper graphs – red vs. black bars), but not in *Nos2^{-/-}* cells (blue vs. green bars). In contrast, *slc40a1* is downregulated in both *Gch1* and *Nos2* deficient cells.

Macrophages were infected with BCG (MOI 1:1)/ IFN γ (10ng/ml) for 24 hours or incubated for 24hrs in the absence of infection (n=4/genotype) followed by RNA extraction. Gene expression was quantified using taqman gene expression assays. Data was analysed by two-tailed T-test * p<0.05.

Whilst detailed analysis of all the cellular processes highlighted in the gene expression analysis is beyond the scope of this manuscript, we have performed additional analyses to validate two of the cellular processes identified as different between *Gch1^{fl/fl}* and *Gch1^{fl/fl}Tie2cre* macrophages - inflammatory activation and cell death/survival.

The differences in inflammatory activation and cell death/survival were detected in both the IPA analysis and the GSEA analysis. We found enhanced expression of IL-6 in infected BH4 deficient macrophages by qRT-PCR in an independent cohort of cells stimulated under the same conditions as the gene array. We also performed ELISA measurement of IL-6 in the supernatant of infected cells after 24 hours and observed enhanced IL-6 secretion by *Gch1^{fl/fl}Tie2cre* macrophages (see Figure, below).

Macrophages were infected with BCG (MOI 1:1)/ IFN γ (10ng/ml) for 24 hours or incubated without infection (n=4/genotype), followed by RNA extraction or ELISA analysis of the supernatant. Gene expression was quantified using taqman gene expression assays. Data was analysed by two-tailed T-test * p<0.05.

To assess whether the pathways associated with cell death or survival translated to a gross difference in cell viability, we established a flow cytometry assay using BCG-GFP to assess cell viability 24hrs after infection. We found no difference in the % cells that remained GFP+ after 24 hours, indicating that our findings were not confounded by differences in BCG uptake by the BMDM. We then quantified the percentage AnnexinV+/7AAD+ dead/late apoptotic cells in both the uninfected and infected cells (by gating on GFP+ vs. GFP- cells) 24hrs following infection. We found no significant difference in cell death/survival between genotypes when cells were infected at an MOI1:1/IFN γ , but a significant increase in AnnexinV+/7AAD+ *Gch1^{fl/fl}Tie2cre* macrophages compared to *Gch1^{fl/fl}* controls, when infected at an MOI 5:1/IFN γ . This experiment shows a difference in the cell death/survival response in BH4-deficient macrophages, in keeping with the alteration in gene expression pathways.

Gch1^{fl/fl}Tie2cre and *Gch1^{fl/fl}* macrophages were stimulated IFN γ (10ng/ml) and BCG (MOI 1:1 or 5:1) or left untreated overnight. At the end of the assay the cells harvested and incubated *Gch1^{fl/fl}Tie2cre* and *Gch1^{fl/fl}* macrophages were stimulated IFN γ (10ng/ml) and BCG (MOI 1:1 or 5:1) or left untreated overnight. At the end of the assay the cells harvested and incubated in buffer containing 7AAD and Annexin V-APC for 15mins at room temperature. Staining was terminated by addition of buffer and cells analysed by flow cytometry immediately. Cell Death was defined as Annexin V⁺ / 7AAD⁺. Infection was confirmed by detection of GFP and early apoptosis (Annexin V⁺ / 7AAD⁻) was assessed in BCG-GFP+ cells infected at MOI 5:1. Data was analysed by 2-way ANOVA * p<0.05 with Bonferroni post-test (n=5/genotype).

Alterations to the cellular cholesterol pathway (Sterol synthesis) was also detected by both IPA and GSEA analysis. Comprehensive analysis of this pathway is beyond the scope of this manuscript as alterations to this pathway can result in altered cholesterol handling and lipid droplet format, altered oxysterol production and alterations to LXR/RXR ligands. The formation of lipid droplets is seen when macrophages are infected by mycobacteria in the presence of IFN γ . We investigated whether this phenomena was still seen in the absence of BH4 using protocols adapted from Knight *et al*, PLOS Pathogens 2018. We observed no lipid droplet formation in the absence of BCG/IFN γ , whereas lipid droplet formation within 24hrs of treatment with BCG/IFN γ was clearly evident in both genotypes. This finding indicates that lipid droplet formation, an important aspect of macrophage function that can impact on mycobacterial persistence, is not grossly compromised in BH4-deficient macrophages, and may be an interesting focus for future studies.

*Gch1^{fl/fl}*Tie2cre and *Gch1^{fl/fl}* macrophages were stimulated with IFN γ (10ng/ml) and BCG (MOI 1:1) or left untreated overnight in the presence of 2% serum as a lipid source. At the end of the assay the cells were fixed and neutral lipids were stained using LipidTox dye and nuclei stained with DAPI. Images are representative of 4 individual macrophage cultures.

These new studies highlight validate the approach used to probe the phenotype of the BH4 deficient macrophages, demonstrating that the pathways highlighted do related to altered cellular function. Determining the precise contribution of all the pathways detected is outside the scope of this manuscript, but we have added this validation data to a new Supplementary Figures 10 and 12 and have discussed this data in the manuscript as follows:

“We validated the presence of genes that show regulation only as a result of BH4 deficiency (*As3mt*) and as a result of loss of both BH4 and iNOS (*Slc40a1*) in our gene array study in an independent cohort of cells, confirming the dichotomy in gene expression between *Gch1* and *Nos2* deficient cells (Supplementary Figure 9).“

and

“However, number of gene signatures were unique to the loss of BH4 (Figure 5C and D). These clusters related fundamental cellular processes such as metabolism and cellular detoxification as well as to processes related to inflammation and infection such as lysosomal function, apoptosis and cell senescence and innate immune. Taken together, the IPA and GSEA data indicate that macrophage *Gch1* and tetrahydrobiopterin are specifically associated with regulation of the innate inflammatory response, cellular metabolism and mitochondrial function, cell detoxification and cell survival. To confirm if the gene expression changes correlated with alterations in cell function we tested whether macrophage inflammatory activation and cell survival were altered, as is indicated by the gene expression analysis. *Gch1^{fl/fl}*Tie2cre macrophages showed reduced IL-6 gene and protein expression in response to BCG/IFN γ , as predicted (Supplementary Figure 10). Under the conditions used for gene expression analysis no alterations in cell viability were seen, under conditions with similar levels of infection of the bulk population (Supplementary Figure 12a). Under conditions of enhanced greater stress, with cells being infected with BCG/IFN γ (MOI 5:1) we observed a decreased macrophage cell survival, associated with increased apoptosis of infected cells, in the absence of *Gch1* again confirming that the gene expression analysis is predictive of alterations in macrophage cell biology. Gene sets related to the Sterol/Cholesterol pathway can have multiple effects on cell lipid handling including altered cholesterol handling and lipid droplet format, altered oxysterol production and alterations to LXR/RXR ligands. We tested the ability of BH4 deficient cells to form lipid droplets in response to LPS/IFN γ , using a recently reported method and droplet formation still occurred in the absence of NO (Supplementary Figure 12b)³². “

4. The authors fail to find a correlation between iNOS expression and BCG growth in human samples (Fig. 6), even though they cite on p. 3 the work from Gomez et al. on the association of iNOS polymorphism with tuberculosis but no further discussion is provided.

We are grateful to the Reviewer for raising this point, and we have updated our manuscript to discuss this interesting question. The SNPs in the *NOS2* gene have indeed been associated with altered susceptibility to disease. Many of the identified SNPs sit in the promoter region and are associated with increased gene transcription and protection from parasitic disease. Gomez et al. demonstrated that the related *NOS2A CCTTT* microsatellite is associated with resistance to TB, as evidenced by a decreased number of individuals being tuberculin skin test positive. These data do indicate that alterations in iNOS biology may have an impact on human susceptibility to tuberculosis.

Our data directly links *NOS2* gene expression and the control of mycobacterial growth acutely and shows that these parameters do not correlate, whereas expression of *GCH1* shows a significant inverse correlation. We speculate that the protective role of *NOS2* in human disease may therefore not be simply due to *NOS2* expression levels, but to the regulation of iNOS activity, something that is not addressed in the earlier studies.

We have addressed this important point in the Discussion, as follows:

“SNPs in the *NOS2* gene have been associated with altered susceptibility to disease, with many identified SNPs sitting in the promoter region and associated with increased gene transcription and associated with protection from TB³. This data does indicate that alterations in iNOS biology has an impact on human susceptibility, however our data would suggest that this may not be a simple association with NO cytotoxicity, but that the relative expression of *NOS2* and *GCH1* is likely to be key.”

5. Several of the data have been published by the same group in McNeill et al., 2015, *Free Radical Biology and Medicine* 79:206–216; this holds through for Fig 1a, c-e and Fig 2a-b and therefore are redundant here. Also the gene expression analysis of *gch-1* versus wildtype mice have been published in the aforementioned paper, revealing 78 genes whose expression was altered between *gch-1* and wild type macrophages. No mentioning of the 2015 dataset is made here; are the 77 genes mentioned in the present manuscript (second paragraph p. 10) overlapping? Is this the same dataset re-analyzed?

We acknowledge that some of the experiments in our previous *FRBM* paper report the response of *Gch1^{fl/fl}Tie2cre* macrophages to cytokine stimulation *in vitro* – but we emphasise that all of the data in the current manuscript reporting the response to mycobacterial infection are completely new, including the gene expression and gene set data.

We believe that it is important to show the effect of mycobacterial infection in *Gch1^{fl/fl}Tie2cre* macrophages *in vitro*, which is critical to the interpretation of the *in vivo* data presented in the manuscript. Furthermore, in Fig 2a-b we present data from BAL cells, (primarily lung macrophages), a cell type that is highly relevant to the *in vivo* phenotype and has not reported previously.

However, we accept that the inclusion of the transgenic locus figure is repetitive, and the initial results text could cover this data in a more concise fashion, given our existing publication. We have removed the transgene diagram to Supplementary Figure 1 to illustrate the location of the primers used in that figure and have substantially altered the initial results text to highlight the novel aspects of our data as follows:

“We generated matched litters of *Gch1^{fl/fl}Tie2cre* and *Gch1^{fl/fl}* mice (hereafter referred to as wildtype), and confirmed that *Gch1^{fl/fl}Tie2cre* mice lack *Gch1* expression in hematopoietic cells *in vivo* showing a widespread haematopoietic deficiency (Figure S1a-c). *In vitro* infection of macrophages with BCG (Pasteur) alone showed minimal induction of *Nos2*, however in the presence of IFN γ infection with BCG caused a robust induction of *Nos2* and accumulation of nitrite, indicating robust iNOS NO-producing activity (Figure 1a). The specificity of nitrite accumulation for iNOS activity was confirmed by infection of *Nos2^{-/-}* BMDM under identical conditions (Figure S2). BCG/IFN γ treatment caused increased cellular BH4 content in wildtype mice, with BH4 levels significantly reduced to near undetectable levels in the *Gch1^{fl/fl}Tie2cre* cells. Loss of BH4 did cause affect

induction of iNOS protein (Figure 1b), but prevented iNOS NO-producing activity measured as nitrite accumulation, arginine-citrulline conversion and direct detection of authentic NO generation by EPR (Figure 1 d-f). In order to test whether iNOS protein had other non NO-producing roles in mycobacterial function the BCG/IFN γ combination was used in all *in vitro* studies to cause robust iNOS expression. These data confirm that whilst iNOS protein is induced in macrophages infected with BCG, in the absence of the co-factor BH4 no nitric oxide is produced, providing a novel system to discriminate iNOS functions reliant specifically on nitric oxide production. “

The gene array data presented in this manuscript is an entirely new and independent dataset produced for this study, in response to mycobacterial infection, and has not been published before. In order to clarify this, we have submitted the earlier gene array dataset to GEO, with accession number GSE112720. A Reviewer log in to this data has been supplied to the journal and is available to the Reviewers.

However, we accept that we did not discuss the earlier dataset in relation to the current study. We have now highlighted the fact that there are striking differences in the expression of key genes such as *IL-6* which is decreased in response to LPS/IFN γ , but significantly upregulated when cells are infected with BCG/IFN γ . This difference highlights that the cellular response to a live pathogen is substantially different to the response to a simple TLR ligand.

We have addressed these important points in the text, as follows:

“These data confirmed the gene expression changes observed in our earlier study of LPS/IFN γ stimulated macrophages, with the identification of similar gene expression changes in both *As3mt* and *Slc40a1*²⁶ and a similar prediction of decreased NRF2-mediated gene expression (Figure 4D). However, it also demonstrated striking divergence regulation of key genes such as IL-6 which showed decreased activation following LPS/IFN γ treatment but enhanced expression following BCG/IFN γ treatment (Supplementary Figure 11 and Dataset 2), indicating exposure to a live pathogen causes a unique pattern of gene regulation that cannot be adequately modelled by the use of other proinflammatory stimuli.”

Other points

1. In Fig 1k and Fig 2f,g the authors highlight the differences between total biopterins in the wild type and mutant cells, but isn't the difference in BH4 levels important? Also, is the difference between wild type and mutants not mainly reflecting the difference in BH4 levels?

We thank the Reviewer for raising this question about the levels of the respective biopterin species. BH4 is the biologically active species for iNOS activity, but is readily oxidised to dihydrobiopterin (BH2), which may be reduced back to BH4 under certain conditions, or can be terminally oxidised to biopterin (B). As described in our previous publications (e.g. references 23, 25, 27), our HPLC system measures all three forms of biopterin, so provides quantification of BH4 and 'total biopterins' (the sum of BH4, BH2 and B), which is commonly used as a surrogate for the biosynthetic activity of biopterin production (by GTPCH), irrespective of the subsequent biochemical fate of BH4. For example, under conditions where tissues cannot be snap frozen immediately significant oxidation can occur. In the case of Figure 2f,g total lung and spleen were homogenised in PBS for CFU assessment. To prioritise CFU assessment antioxidants were not included in the homogenisation buffer to chemically protect BH4. Similarly in the case of Figure 1k the cell free lavage fluid contained

only oxidised biopterins, presumably released from cells, the accompanying BH4 content of the cell pellets as presented in Figure 1l.

We have now referred to this directly in the text as follows:

“Analysis of peritoneal lavage fluid in wildtype mice revealed that zymosan caused significant induction of the iNOS and tetrahydrobiopterin pathway, measured by total biopterins and nitrite/nitrate levels (NOx), stable oxidized extracellular endproducts of nitric oxide production *in vivo* (Figure 1j,k). Total biopterins (BH4, dihydrobiopterin and biopterin) rather than BH4 were assessed due to the oxidizing nature of extracellular environment causing oxidative loss of BH4 to the oxidized forms dihydrobiopterin and biopterin.”

And

“Total biopterin levels rather than BH4 alone were assessed due to the oxidizing nature of the homogenisation process required for BCG CFU assessment. “

2. While for BCG infections lung CFU's are similar for wild type and *gch-1*^{-/-} mice and spleen CFU's reduced in the *gch-1*^{-/-} mice, this is reversed for the *M. tuberculosis* infection shown in Fig. 3, without providing any explanation.

We are grateful to the Reviewer for raising this point, and we accept that we did not address the potential differences between lung and spleen involvement, and how these may differ when different mycobacterial strains are used in models of primary respiratory infection.

The BCG infection data showed some evidence of reduced susceptibility in *Gch1*^{-/-} mice, with a trend to reduced CFU counts in the lung and significantly reduced extrapulmonary dissemination to the spleen, compared to WT mice, 4 weeks after infection. A similar lack of increased susceptibility, compared to the iNOS^{-/-} mice, is seen in the lung in the early (24 day) timepoint shown in Figure 3C, although with this more virulent infection the effect on extrapulmonary dissemination is not seen. Infection with the more virulent *Mtb* is a more robust assessment of susceptibility and may overcome the relative reduced susceptibility seen using the less virulent BCG infection. Analysis of BH4-deficient mice 6 weeks following infection revealed a reproducible decrease in CFU in the lungs of the BH4 deficient mice indicating improved control at this timepoint. A 6 week infection study was not performed using BCG so cannot be compared.

We have altered the text to comment on this observation as below:

“Whilst the reduced extrapulmonary spread of infection, seen when mice were infected with BCG, was not maintained when *Gch1*^{fl/fl}*Tie2*^{cre} mice are exposed to the more virulent *M.tb* infection, this stringent test of susceptibility shows a clear reduced susceptibility in the lung at a later 6-week timepoint. ”

3. All blots should be labeled with the appropriate molecular weight markers.

We are grateful to the Reviewer for highlighting this omission, which has now been corrected.

4. The mouse data shown reflect different group sizes, especially in Figs 2c,d (wild type mice n=10; mutant mice n=7 (2c) and n=6 (2d)) without any explanation as to why these mice numbers are different.

Our experiments use age-matched sex-matched littermate animals. As a consequence, the number of each genotype available for each experiment is not always identical. In addition, we always undertake robust re-genotyping protocols following termination of the experiment to confirm the genotype of all experimental animals. In the case of the data presented in Figure 2 re-genotyping of the experiment identified that one of the animals initially identified as *Gch1^{fl/fl}Tie2cre* was in fact a wildtype, increasing the size of the wildtype group. Additionally, one spleen CFU sample was excluded as an outlier following statistical testing using the Grubb's outlier test. Similar testing of all other groups revealed no further statistical outliers. This has been highlighted in the methods.

“CFU studies were analyzed by Man Whitney test, with statistical outliers identified using Grubb's test and removed from the dataset. In this case a single value was removed from the spleen CFU values in Figure 2. “

5. Page 11 line 7: Figure 7 should be Fig 5.

This has been corrected.

Reviewers' comments:

Reviewer #2 (Remarks to the Author):

The authors have carefully addressed all concerns, and I congratulate the authors with a great piece of work. I have just one comment and one minor typing issue:

Comment

- Comment: The authors now far better document and describe the modulation of ROS upon gch1 deletion. However, the discussion section lacks any clear statement as to whether ROS could compensate for the loss of iNOS in these mice. Their previous version contained a statement to that issue ('...the contrast between Nos2-/- cells, that lack all aspects of iNOS activity (both NO and ROS-mediated), vs. Gch1fl/flTie2cre cells (that) lack that NO production, but have maintained and potentially enhanced ROS production...') that I could not find in the revised manuscript, but which would be good to add back.

Typing issue:

- On page 3 (Introduction, line 63) the authors write M.Bovis Bacillus Calmette Guerin, which should be M. bovis Bacillus Calmette Guerin

Reviewer #3 (Remarks to the Author):

I have assessed the GSEA and in my opinion there are important missing details, which make the interpretation of the results (and their actual relevance) very difficult. I detail these critical points below.

(1) The Authors used "GSEAPreranked module using the GO gene sets (C5)", which contains genes annotated by the same GO term and divided into three sub-collections based on GO ontologies: BP, CC, MF. As the Authors aimed to "evaluate the cellular functions" I also suggest to use a wide, more comprehensive, set of gene set, including: Gene sets curated from various sources such as online pathway databases, the biomedical literature, and knowledge of domain experts; Canonical pathways; Gene sets derived from the BioCarta pathway database; KEGG: KEGG gene sets. This analysis will be more informative than focusing on GO terms only.

(2) "We compared the two network maps to determine which gene sets were shared and which were unique to the loss of gch1 (Figure 5C & D)."
This procedure is not explained in the Methods and as is very unclear. More details are required for the reader to understand exactly how the shared or unique networks or sub-network were identified. Which method was used to compare the networks? Does this account for the network topology? Is there a quantitative assessment for the network sharing/uniqueness?

(3) There are methodological points from the legend of Fig. 5 that are unclear and/or incorrect, these require to be addressed (or adequately clarified).

3a. "Gene sets passing FDR $p < 0.1$ with an overlap coefficient 0.5 were used to plot an Enrichment Map in Cytoscape." How is the overlap calculated? overlap between what and what? How was the threshold 0.5 chosen? Are the results affected by this choice?

3b. The "FDR $p < 0.1$ " does not make much sense as the FDR (false discovery rate) is not a P-value and the statistical interpretations of FDR and P-value are very different. This is probably just $FDR < 0.1$, which means accepting 10% of false discoveries. In fairness, this remark and correction should apply here and in several other parts of the manuscript where "FDR p" is wrongly used.

Reviewers' comments:

Reviewer #2 (Remarks to the Author):

The authors have carefully addressed all concerns, and I congratulate the authors with a great piece of work. I have just one comment and one minor typing issue:

Comment

- Comment: The authors now far better document and describe the modulation of ROS upon *gch1* deletion. However, the discussion section lacks any clear statement as to whether ROS could compensate for the loss of iNOS in these mice. Their previous version contained a statement to that issue ('...the contrast between *Nos2*^{-/-} cells, that lack all aspects of iNOS activity (both NO and ROS-mediated), vs. *Gch1**fl/flTie2cre* cells (that) lack that NO production, but have maintained and potentially enhanced ROS production...') that I could not find in the revised manuscript, but which would be good to add back.

We thank the Reviewer for their positive comments about our revised manuscript, and their valuable guidance in directing the production of this improved version. We have now added back the section on NO vs. ROS-mediated biology into the discussion, as below (new text highlighted in red):

*"While the cytotoxic capacity of the cell was assumed to then reside in the ability to produce RNI and ROS species, increasingly evidence suggests this is not the case. The role of ROS in direct mycobacterial control is as controversial as the role of nitric oxide. The loss of ROS production in *p47*^{PHOX}^{-/-} mice resulted in only a transient decrease in bacterial control, and human chronic granulomatous disease patients, who lack myeloid cell ROS production, do not show increased susceptibility to TB^{46, 47}. An interpretation of our findings is that *Nos2*^{-/-} cells lack all aspects of iNOS activity (both NO and ROS-mediated), whereas *Gch1**fl/flTie2cre* cells lack NO production but have preserved ROS production, that may mediate mycobacterial killing. A direct effect of ROS on mycobacterial killing seems less likely, given both the complicated pattern of altered ROS production in the absence of BH4 and equivocal evidence for mycobacterial killing by ROS. Nevertheless, altered ROS production may exert important effects through altered downstream redox signalling ."*

Typing issue:

- On page 3 (Introduction, line 63) the authors write M.Bovis Bacillus Calmette Guerin, which should be M. bovis Bacillus Calmette Guerin

We have corrected this typo.

Reviewer #3 (Remarks to the Author):

I have assessed the GSEA and in my opinion there are important missing details, which make the interpretation of the results (and their actual relevance) very difficult. I detail these critical points below.

We thank the Reviewer for their careful assessment of the GSEA and for identifying omissions and points of clarification we have overlooked. We have now added a further round of GSEA analysis using the additional gene set sources suggested, and have clarified our methods – including addition of Venn Diagrams to highlight the overlap between the two different genotypes analysed.

(1) The Authors used “GSEAPreranked module using the GO gene sets (C5)”, which contains genes annotated by the same GO term and divided into three sub-collections based on GO ontologies: BP, CC, MF. As the Authors aimed to “evaluate the cellular functions” I also suggest to use a wide, more comprehensive, set of gene set, including: Gene sets curated from various sources such as online pathway databases, the biomedical literature, and knowledge of domain experts; Canonical pathways; Gene sets derived from the BioCarta pathway database; KEGG: KEGG gene sets. This analysis will be more informative than focusing on GO terms only.

We thank the Reviewer for these suggestions to include additional curated gene sets in order to increase the biological relevance and validity of the findings. We had initially selected our analysis of GO gene sets relating to biological processes (C5 BP), because GSEA based on GO terms is particularly suited for network analysis using Enrichment Map, which takes advantage of the large number of overlapping gene sets to produce clusters of related gene sets (in this case relating to core cellular processes). We found this a helpful way of drawing conclusions from our GSEA analysis. However, we entirely accept that the confidence in this analysis would be increased by including the new analyses suggested by the Reviewer.

We have now included our GSEA analysis using the ‘Curated Pathways – Canonical Pathways’ gene sets, encompassing Reactome, KEGG, PID and Biocarta gene sets. Analysis of the shared *Gch1* and *Nos2* vs. *Gch1* unique gene sets reveals that some gene sets, such as those related to ribosome function, are conserved in both responses to infection, as are elements of the enhanced inflammatory response, such as cytokine-cytokine receptor interactions. This new analysis highlights gene sets relating to the core cellular processes identified in our IPA and GO term based analysis, including lysosomal function, chemokine signaling, response to DNA damage and lipid metabolism, confirming our existing analytical approach. However, this new analysis has more specifically highlighted pathways such as p53 and mitochondrial protein import within the broader cellular functions that show altered regulation.

We would also highlight that in response to the original comments by Reviewer 2, we have included in the manuscript biological validation of selected candidate pathways identified in our genetic analyses, e.g inflammatory cytokine production and cell survival/apoptosis.

Given the importance of the new GSEA analysis, we have included these findings in a new supplementary figure, shown below.

Supplementary Figure 12: Gene set enrichment using GO term functional annotations identifies shared gene expression patterns in BCG/IFN γ treated macrophages. Enriched gene sets in BCG infected macrophages, identified by Preranked GSEA (MSigDb, C2 CP annotations set). To determine how far regulated geneset were shared between the *Gch1* and *Nos2* knockout macrophages or unique to the loss of *Gch1* the gene set lists passing $p < 0.005$, FDR $q < 0.1$ in both analyses were compared (a) and key genesets were are highlighted in the tables.

We have updated the corresponding text in the manuscript, as follows:

“To more broadly evaluate the cellular functions mediating enhanced control of BCG infection in Gch1-deficient macrophages using the entire dataset, we performed unsupervised Gene Set Enrichment Analysis (GSEA) to identify coordinated regulation of annotated groups of genes, implicated in defined cellular processes 31. First, we formed lists of all genes ranked by difference from the relevant wildtype control, then performed Preranked GSEA using the GO term-defined gene sets relating to biological processes (MSigDB C5 BP). Network analysis of the most significantly altered gene sets for each analysis was performed, using Enrichment Map³¹. This analysis was used to

visualize clusters of overlapping gene sets within the GSEA data for infected Gch1 and Nos2 deleted macrophages vs their wildtype controls indicating biological processes likely to be altered (Figure 5A and B, Dataset 3). We compared the lists of significantly altered gene sets to identify which were shared between genotypes and which were unique to the loss of gch1 (Supplementary Figure 11A). These new lists of shared and unique gene sets were again used to visualize gene set clusters using Enrichment Map (Figure 5C and Supplementary Figure 11B). The majority of gene sets relating to DNA repair, protein catabolism, vesicular trafficking and ribosome activity were found to be common to the loss of either Gch1 or Nos2 (Supplementary Figure 11B and Dataset 3). However, a number of gene signatures were unique to the loss of Gch1 (Figure 5C and D). These gene sets were related to fundamental cellular processes such as metabolism and cellular detoxification as well as to processes related to inflammation and infection such as lysosomal function, apoptosis and cell senescence and innate immune. To confirm the biological relevance and validity of these findings, we performed a second round of GSEA using gene sets relating to canonical pathways and curated datasets (MSigDB C2 CP). This analysis identified a similar large number of significantly regulated pathways in the Nos2 dataset, but again identified a smaller number of gene sets that were either shared by both genotypes or unique to the loss of Gch1 (Supplementary Figure 12 and Dataset 3). Gch1 unique gene sets again included those related to lysosome function, inflammation, DNA damage response including the p53 pathway) and lipid and glucose metabolism. “

(2) “We compared the two network maps to determine which gene sets were shared and which were unique to the loss of gch1 (Figure 5C & D).”

This procedure is not explained in the Methods and as is very unclear. More details are required for the reader to understand exactly how the shared or unique networks or sub-network were identified. Which method was used to compare the networks? Does this account for the network topology? Is there a quantitative assessment for the network sharing/uniqueness?

We accept that our description here was not sufficiently clear. The networks were plotted in Cytoscape using Enrichment Map, but the identification of shared and unique gene sets was made using a simple subtraction of the Nos2 and Gch1 -regulated gene sets. This new subset of Gch1 unique gene sets was then used to produce a new Enrichment Map.

In order to make this procedure clearer, we have added an explanatory Venn Diagram in the supplementary data, and we have clarified our description of the data in the results, legend and online methods sections. We have also updated the GSEA Dataset (Dataset 3) submitted with the manuscript in to a simpler format with tabs for unique or shared gene sets for both GSEA analyses and the NES, NOM p value and FDR q value for each gene set passing our defined significance threshold.

Supplementary Figure 11: Gene set enrichment using GO term functional annotations identifies shared gene expression patterns in BCG/IFN γ treated macrophages. Enriched gene sets in BCG infected macrophages, identified by GSEA (MSigDb, C5 GO BP annotations set). To determine how far regulated gene functions were shared between the *Gch1* and *Nos2* knockout macrophages the gene sets passing $p < 0.005$, FDR $q < 0.1$ in both analyses were compared (a) (excerpt of legend).

(3) There are methodological points from the legend of Fig. 5 that are unclear and/or incorrect, these require to be addressed (or adequately clarified).

3a. “Gene sets passing FDR $p < 0.1$ with an overlap coefficient 0.5 were used to plot an Enrichment Map in Cytoscape.” How is the overlap calculated? overlap between what and what? How was the threshold 0.5 chosen? Are the results affected by this choice?

We thank the Reviewer for identifying the error in the Figure 5 legend. We used the Enrichment Map plugin for Cytoscape. In line with the guide for using this analysis, we used a moderately conservative statistical cut off for geneset permutations of $p < 0.005$ and FDR $q < 0.1$. The overlap coefficient between the two genesets (A and B) used to is calculated within the software as:

$$\text{Overlap Coefficient} = [\text{size of (A intersect B)}] / [\text{size of (minimum(A , B))}]$$

0.5 is moderately conservative, and is recommended by the originator lab for most standard analyses and is used to determine linked gene set nodes within the network and plot edge thickness.

We have also corrected the oversight, which had removed the references for the Enrichment Map plugin analysis to only the Online Methods, making our precise process less clear.

We have updated the legend to Figure 5 as follows:

“Figure 5: Gene ontology (GO) enrichment analysis identifies shared and unique gene expression patterns in Gch1 and Nos2 deficient BCG/IFN γ treated macrophages. Gene expression data from both Gch1 and Nos2 deficient macrophages were analyzed for GO term (biological processes) enrichment by Preranked gene-set enrichment analysis (GSEA) vs their wildtype control. The result was visualized on a network of gene sets (nodes) connected by their similarity (edges), compiled in Cytoscape using the Enrichment Map plug in. Node size represents the gene-set size and edge thickness represents the degree of overlap between neighboring gene sets. The GSEA output files were given analysed with a statistical cutoff set at $p < 0.005$ and FDR $q < 0.1$. The individual Enrichment Maps were plotted in Cytoscape for infected (a) Gch1 and (b) Nos2 macrophages and clusters were annotated by hand. To determine if regulated gene sets were shared between the Gch1 and Nos2 knockout macrophages the significantly regulated gene sets were compared and a subset of gene sets unique to Gch1 deficient macrophages was created, this subset (c) was plotted using Enrichment Map and the clusters annotated. (d) A table summarising the GO-term clusters, with exemplar GO terms.”

And in the Methods section:

“For Gene Set Enrichment Analysis (GSEA) preranked lists for the effect of genotype within infected samples were created. GenePattern v 3.9.10 (Broad Institute) was used and lists were subjected to analysis the GSEAPreranked module using the GO biological processes gene sets (C5 BP) and Curated Pathways (C2 CP) from the Molecular Signature Database v6.0 (MSigDB)^{12, 13, 14}. Gene Sets with a FDR $q < 0.1$ and p value $p < 0.005$ were considered significant in this analysis and used in further analyses. Network analysis of significantly regulated gene sets ($p < 0.005$ and FDR $q < 0.1$) was performed using Cytoscape v3.5.1 and the Enrichment map plugin^{15, 16, 17}, the Enrichment Map default overlap coefficient of 0.5 was used, as described in¹⁷. To create a list of gene sets unique to the loss of Gch1 any genes passing $p < 0.005$ and FDR $q < 0.1$ present in both Gch1 and Nos2 deficient macrophage analyses were removed from the Gch1 list and the remaining gene sets replotted in Cytoscape..”

3b. The “FDR $p < 0.1$ ” does not make much sense as the FDR (false discovery rate) is not a P-value and the statistical interpretations of FDR and P-value are very different. This is probably just FDR < 0.1 , which means accepting 10% of false discoveries. In fairness, this remark and correction should apply here and in several other parts of the manuscript where “FDR p ” is wrongly used.

We thank the Reviewer for highlighting these typos, the Figure 4 legend should read:

“ Genes were determined as significantly regulated by genotype by comparison to their relevant wildtype control with $p < 0.05$ (adjusted for multiple testing). ”

Similarly, we have corrected the threshold we used to define our gene sets of interest as $p < 0.005$ and FDR $q < 0.1$ in the text, referring to gene set enrichment thresholds used.

REVIEWERS' COMMENTS:

Reviewer #2 (Remarks to the Author):

The authors have fully addressed all my concerns.

Reviewer #3 (Remarks to the Author):

The Authors have addressed my concerns on the GSEA.

I am happy to see that the new suggested analyses helped the Authors to increase the biological relevance and validity of the findings presented, and improve the manuscript's readability.

Overall, this is a nice piece of work. Congratulations.

NCOMMS-17-29198B: Regulation of Mycobacterial Infection by Macrophage *Gch1* and Tetrahydrobiopterin

We would like to thank the reviewers and editors for the handling and critical reading of this Manuscript. We have modified the manuscript in response to the editorial formatting requests. The changes have been made using the track change function of Microsoft Word.

Reviewers' comments:

Reviewer #2 (Remarks to the Author):

The authors have fully addressed all my concerns.

Reviewer #3 (Remarks to the Author):

The Authors have addressed my concerns on the GSEA.

I am happy to see that the new suggested analyses helped the Authors to increase the biological relevance and validity of the findings presented, and improve the manuscript's readability.

Overall, this is a nice piece of work. Congratulations.

We thank the reviewers for their constructive comments during the revision of this manuscript and appreciate their enthusiasm for this work through the review process.